# REPORT

# Mitochondrially tethered Mmm1 can function as a sole lipid transporter at ER–mitochondria contacts

Christian Covill-Cooke[1]*⊙, Takashi Hirashima[2,3]*⊙, Shin Kawano[2]*⊙, Joe Ganellin[1]⊙, Andrew Moody[1]⊙, Sabine N.S. van Schie[4]⊙, Arun T. John Peter[4]⊙, Chika Horie Saito[2]⊙, Toshiya Endo[2,3]⊙, and Benoît Kornmann[1]⊙

**Yeast mitochondria receive the majority of their lipids from the ER via the heterotetrameric ERMES lipid transport complex. This complex is thought to establish a lipid-transporting bridge of fixed composition spanning the space between both organelles. Intriguingly, however, some of the lipid-transporting components of the complex can be replaced by an artificial ER–mitochondria tether without lipid transport activity, questioning ERMES' relevance in lipid transport. Here, we show that Mmm1, one of the four ERMES subunits, alone is sufficient to support ERMES function when it is artificially tethered to mitochondria, provided its lipid-binding domain is intact. Combined with our previous finding that the absence of Mdm12 and Mdm34 can be rescued by the presence of Mmm1 and the artificial tethering protein ChiMERA, our results suggest that Mmm1 can act as the sole lipid transporter at the ER–mitochondrial contact sites, provided that Mdm10 is present, even in the absence of the other two subunits. Thus, our work reconciles ERMES' importance in lipid transport with the fact that the lipid transport activity of some of its components is not strictly necessary for function.**

## Introduction

The transport of lipids from their site of synthesis—predominantly the ER—to the mitochondrial membranes is essential for the existence of mitochondria and, therefore, eukaryotic life (Tatsuta and Langer, 2017; Tamura et al., 2020). In yeast, the transport of lipids from the ER to mitochondria is thought to occur predominantly through the ERMES complex, a hetero-tetrameric complex of Mmm1, Mdm12, Mdm34, and Mdm10 (Kornmann et al., 2009; Kornmann, 2020; John Peter et al., 2022). The ability of ERMES to traffic lipids is bestowed by lipid-transfer synaptotagmin-like mitochondrial lipid-binding protein (SMP) domains found in three of the members of the complex (Mmm1, Mdm12, and Mdm34). SMP domains form hydrophobic pockets that can shelter lipids from the aqueous cytosol, thus catalyzing interorganelle lipid exchange (AhYoung et al., 2015; Ah Young et al., 2017; Kopec et al., 2010; Kawano et al., 2018; Jeong et al., 2016; Jeong et al., 2017).

Important questions remain unaddressed regarding the biology of the ERMES complex. First, how does it transfer lipids at the mechanistic level? A crystal structure of an Mmm1-Mdm12 heterotetramer (homodimer of heterodimers) has been described (Jeong et al., 2017). Given the size of this heterotetramer,

it has been proposed to shuttle lipids between both membranes using flexible linkers in Mmm1 and/or Mdm34. By contrast, biochemical analyses (Ellenrieder et al., 2016) as well as cryo-electron tomography observations with the aid of structural prediction fitting (Wozny et al., 2023) indicate that the SMP domain of Mmm1 (anchored in the ER membrane) connects to that of Mdm12, which itself binds to Mdm34 to bridge over to Mdm10, embedded in the outer mitochondrial membrane (OMM). In this model, a linear and compositionally rigid Mmm1-Mdm12-Mdm34-Mdm10 heterotetramer constitutes a hydrophobic conduit for lipid transfer between the two membranes (Fig. 1 A). Yet, because the folded domains of Mmm1, Mdm12, and Mdm34 are similar in shape and size, at the resolution of the tomograms, one cannot exclude the existence of a variety of complexes with different subunit compositions, like Mmm1–Mdm12–Mdm12–Mdm10 or Mmm1–Mdm34–Mdm34–Mdm10, for instance.

Despite structural evidence for the involvement of ERMES in mitochondrial lipid supply, two bits of genetic evidence contradict such a function. First, while deletion of any of its members leads to the disappearance of ERMES structures, slow

---

[1]Department of Biochemistry, University of Oxford, Oxford, UK;   [2]Faculty of Life Sciences, Kyoto Sangyo University, Kyoto, Japan;   [3]Institute for Protein Dynamics, Kyoto Sangyo University, Kyoto, Japan;   [4]Institute of Biochemistry, ETH Zurich, Zurich, Switzerland.

*C. Covill-Cooke, T. Hirashima, and S. Kawano contributed equally to this paper.   Correspondence to Benoît Kornmann: benoit.kornmann@bioch.ox.ac.uk;   Toshiya Endo: tendo@cc.kyoto-su.ac.jp

A.T. John Peter's current affiliation is Life Science Institute, University of British Columbia, Vancouver, Canada.

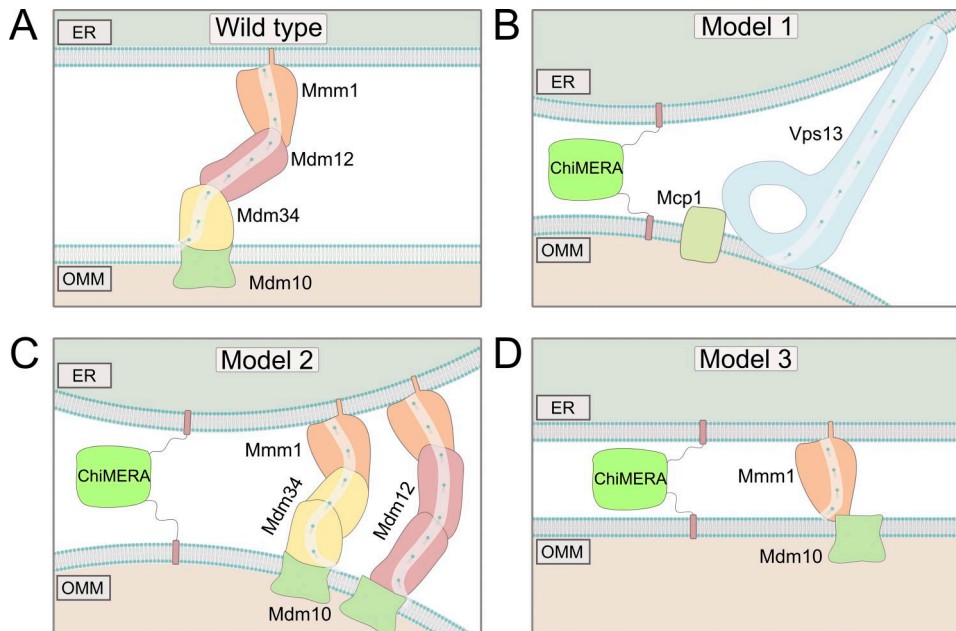

Figure 1. **Models for ChiMERA rescue of ERMES depletion. (A)** Model of the heterotetrameric ERMES complex. **(B)** Model 1: The lipid transporter Vps13 is recruited to artificial ER–mitochondria contacts. **(C)** Model 2: Mdm12 and Mdm34 are essential for tethering but redundant with one another in lipid transport function. **(D)** Model 3: Mmm1 can function as the sole LTP if there is sufficient external tethering. Mdm10 supports lipid transfer by providing an anchoring point for Mmm1 to mitochondria.

growth, and mitochondrial malfunction, it does not entirely prevent cell growth nor mitochondrial membrane biogenesis in budding yeast (Kornmann et al., 2009; Kornmann, 2020; Burgess et al., 1994; Boldogh et al., 2003; Sogo and Yaffe, 1994; Berger et al., 1997), indicating that ERMES is not strictly necessary to provide lipids to mitochondria.

Second, the absence of the ERMES complex can be suppressed by the expression of ChiMERA—a synthetic fusion protein that artificially tethers the ER to the mitochondria. More specifically, ChiMERA expression rescues the growth defect of *mdm12* and *mdm34* single mutants although it fails to rescue that of *mmm1* and *mdm10* mutants (Kornmann et al., 2009). Crucially, ChiMERA lacks lipid transport activity, at odds with the idea that ERMES is a lipid transport catalyst (Kornmann et al., 2009). While the first discrepancy has been explained by the existence of redundant lipid transport pathways, most notably involving the lipid transport protein (LTP) Vps13 and its mitochondrial adaptor Mcp1 that support lipid transport between the endomembrane system and mitochondria, and promote cell growth in the absence of ERMES (John Peter et al., 2017; John Peter et al., 2022; Lang et al., 2015; Park et al., 2016; Tan et al., 2013), the second remains unexplained.

Here, we build upon the less well acknowledged observation that, while ChiMERA expression rescues the growth defect of *mdm12* and *mdm34* single mutants, it fails to rescue that of *mmm1* and *mdm10* mutants (Kornmann et al., 2009), and we explore three potential models: (1) ChiMERA-induced tethering might exert its rescuing effect by allowing Vps13 recruitment to artificial ER–mitochondria contact, similar to Mcp1 overexpression, leading to rescue of ERMES mutants (Fig. 1 B); (2) While the presence of both Mdm12 and Mdm34 might be necessary for

ERMES tethering function, both proteins might be redundant with one another for lipid transfer in the presence of ChiMERA, as could be expected if Mmm1–Mdm12–Mdm12–Mdm10 or Mmm1–Mdm34–Mdm34–Mdm10 complexes of lower affinities existed (Fig. 1 C); (3) Mmm1 might be able to serve as the sole LTP at ER–mitochondria contact sites established by ChiMERA-mediated tethering (Fig. 1 D). Distinguishing between these three models has important implications for the mechanism of ERMES-mediated lipid transport.

## Results and discussion

### Vps13 localizes at ER–mitochondria contacts but is dispensable for ChiMERA-mediated ERMES rescue

Model 1 states that ChiMERA expression recruits Vps13 to artificial ER–mitochondria contacts (Fig. 2 A). Although we have previously suggested that Vps13[D716H] rescued ERMES by connecting mitochondria to vacuoles, (John Peter et al., 2017), we found that Vps13's vacuolar adaptor, Ypt35, is not necessary for Vps13-mediated ERMES rescue because transposons can be inserted into the *YPT35* locus (Fig. S1 A) (Bean et al., 2018; Michel et al., 2017). Ypt35 was not required for the recruitment of Vps13 to vacuolar and mitochondrial patches upon Vps39 and Mcp1 co-overexpression, as well (Fig. S1 B). This suggests that Vps13 likely binds promiscuously to membrane apposition sites upon Mcp1-mediated recruitment. In the absence of Vps39 overexpression, Vps13^GFP did not show uniform localization on the mitochondrial surface. Instead, Vps13^GFP appeared to form distinct bright puncta (Fig. S1 C). These bright foci of Vps13 in Mcp1-overexpressing cells frequently colocalized with ERMES, at a rate significantly higher than expected by chance

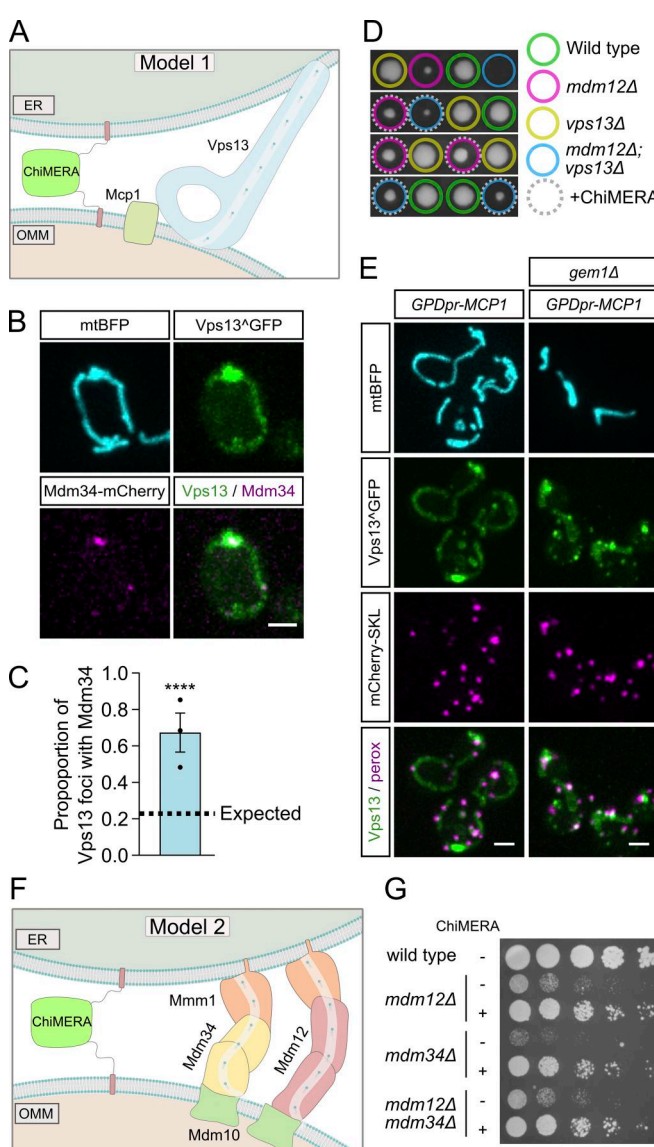

Figure 2. **Vps13 and joint loss of Mdm12-Mdm34 are not required for ERMES rescue by ChiMERA. (A)** Schematic showing the model in which Vps13-Mcp1 can transfer lipids at ChiMERA-induced ER–mitochondria contact sites. **(B)** Representative images of Vps13^GFP colocalizing with Mdm34-mCherry upon Mcp1 overexpression. **(C)** Quantification of the proportion of Mcp1 overexpression-dependent Vps13^GFP foci expected and observed to co-localize with Mdm34-mCherry. Statistical significance was quantified with a chi-squared test. The expected value is probability a Vps13^GFP foci would colocalize by random chance. Error bars indicate ± standard error of the mean. Statistical N is three independent experiments. **** denotes $P < 0.0001$. **(D)** Representative tetrads from the sporulation of *MDM12/mdm12Δ VPS13/vps13Δ* diploids expressing ChiMERA. **(E)** Representative images of Vps13^GFP subcellular localization in comparison with mitochondria (mtBFP) and peroxisomes (mCherry-SKL) in Mcp1 overexpression conditions, both with and without Gem1. **(F)** Schematic of Model 2, whereby Mdm12 and Mdm34 are functionally redundant with one another upon ChiMERA-induced tethering of the ER to the mitochondria. **(G)** Representative spot assay of *mdm12Δ, mdm34Δ*, and *mdm12Δmdm34Δ* both with and without ChiMERA expression. Scale bars are 2 μm.

(Fig. 2, B and C; see Material and methods), indicating a novel localization of Vps13 at ER–mitochondria contact sites. These mitochondrial Vps13 foci are not observed in the absence of Mcp1 overexpression (Fig. S1 C).

The localization of Vps13 to ER–mitochondria contact sites makes Vps13 a strong candidate for a lipid-transfer protein at these sites. If ChiMERA rescued ERMES deficiency through promoting lipid transfer by Vps13 at ER–mitochondria contact sites, ChiMERA expression would be expected to be ineffective in *vps13Δ* cells. To test this idea, we sporulated and tetrad-dissected an *MDM12/mdm12Δ; VPS13/vps13Δ* heterozygous diploid strain, transformed with a ChiMERA-expressing plasmid. As ERMES deficiency is synthetically lethal with *vps13Δ* (Lang et al., 2015), we expect that, if Vps13 was required for ChiMERA-mediated rescue, *mdm12Δ/vps13Δ* cells would be inviable irrespective of ChiMERA expression. While meiotic tetrad dissection showed that *mdm12/vps13* double knockout cells were not viable (Fig. 2 D), expression of ChiMERA led to a consistent growth restoration in these double knockout cells. Therefore, ChiMERA-induced rescue of ERMES deficiency does not require Vps13-dependent lipid transport between the ER and mitochondria, ruling out model 1.

Instead of playing a role in the ERMES rescue pathway, the bright Vps13 foci observed upon Mcp1 overexpression appeared to be mitochondria-derived compartments (MDCs), mysterious outer membrane–enriched organelle protrusions involved in quality control, longevity, and resistance to amino acid stress (Schuler et al., 2021). Like MDCs, Vps13-foci formed at or in proximity to ERMES foci, sometimes exhibited a doughnut shape (Fig. S1 D), and shared the mitochondrial puncta enrichment observed with the bona fide MDC marker, Tom70 (Fig. S1 E). Development of these foci was strictly dependent on Gem1, a calcium-binding GTPase (Frederick et al., 2004), and facultative component of the ERMES complex (Kornmann et al., 2011; Covill-Cooke et al., 2024; English et al., 2020)—Gem1 deletion abrogated the punctate localization of Vps13, resulting in a diffuse mitochondrial signal alongside some enrichment in foci that colocalized with peroxisomes (Fig. 2 E and Fig. S2). Vps13 localization to peroxisomes has been previously observed, but its function is not understood (John Peter et al., 2017). We therefore conclude that Vps13's localization to ER–mitochondrial contact sites is more likely associated with MDC biology and unconnected to the ChiMERA-induced rescue of ERMES deficiency.

## Redundancy between Mdm12 and Mdm34 does not explain ChiMERA-mediated ERMES deficiency rescue

The expression of ChiMERA can rescue the growth defect observed in the single deletion of either *mdm12* or *mdm34* (Kornmann et al., 2009). Model 2 implies that ER–mitochondria tethering by ERMES requires all four members of the complex but that the functions of Mdm12 and Mdm34 in lipid transfer are redundant with one another (Fig. 2 F), which becomes apparent when artificial tethering is provided by ChiMERA. Thus, one would expect that ChiMERA could rescue growth of *mdm12Δ* or *mdm34Δ* single mutants but not of *mdm12Δ/mdm34Δ* double mutants. To test this hypothesis, *MDM12/mdm12Δ; MDM34/mdm34Δ* heterozygous diploids were transformed with a ChiMERA-expressing plasmid, and cell growth was examined following sporulation and tetrad dissection. ChiMERA expression elicited a similar rescue in *mdm12Δ/mdm34Δ* double knockout cells to that obtained in single deletion yeast (Fig. 2 G

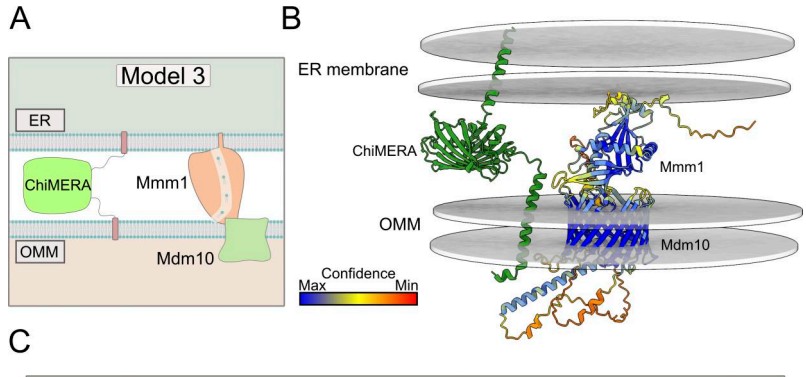

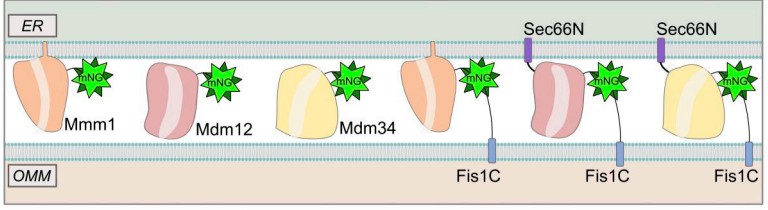

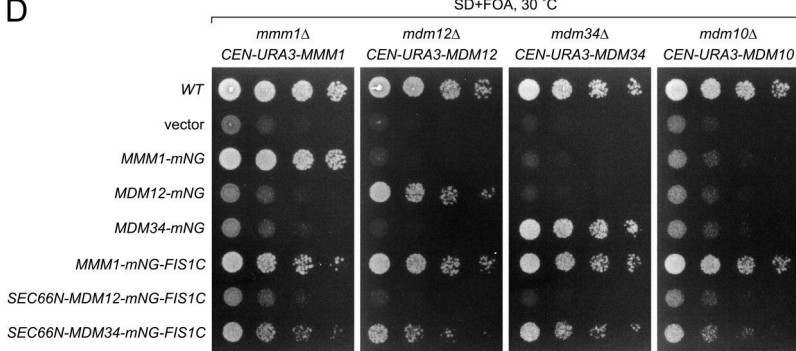

Figure 3. **Artificial tethering of Mmm1 to mitochondria can rescue loss of ERMES. (A)** Schematic of model 3 showing that a Mmm1–Mdm10 dimer is functional upon additional tethering. **(B)** AlphaFold 3 prediction of Mmm1 and Mdm10 dimer. ChiMERA was added manually along with the ER and OMM. **(C)** Schematic of constructs used in D. **(D)** Spot assay of single ERMES deletion yeast expressing artificially tethered ERMES members.

and Fig. S3 A). We therefore conclude that Mdm12 and Mdm34 are not redundant with one another for lipid transport, ruling out model 2.

## The SMP domain of Mmm1 is sufficient for lipid transfer to mitochondria

Contrary to deficiencies in Mdm12, Mdm34, and the combination thereof, deficiencies in Mmm1 and Mdm10 are poorly, if at all, rescued by ChiMERA expression (Cortés Sanchón et al., 2021; Kornmann et al., 2009), indicating that these components play a special role beyond bridging the two membranes, a function that ChiMERA can fulfil. We therefore hypothesized that Mmm1 and Mdm10 might be able to form a dimer that is functional in lipid transfer (model 3, Fig. 3 A). In fact, AlphaFold 3 predicts an Mmm1–Mdm10 heterodimer in an orientation compatible with lipid transfer between two membranes and of an appropriate size to fit within the close apposition between the ER and mitochondria, induced by ChiMERA-mediated tethering (Fig. 3 B and Fig. S3 D). Since Mdm10 is devoid of a lipid transport domain, Mmm1—which itself is ER-anchored—appears to be sufficient for interorganelle lipid transport, with Mdm10 serving as a recruitment factor on the mitochondrion.

We thus asked whether Mmm1 artificially tethered to mitochondria could functionally replace the core subunits of the ERMES complex. To achieve this, we expressed Mmm1 fused to

the fluorescent mNeonGreen (mNG) protein with a C-terminal OMM-targeting transmembrane domain (Fis1-TM, hereafter Mmm1-mNG-Fis1C) (Fig. 3 C). Interestingly, Mmm1-mNG-Fis1C could rescue not only the growth defects of an mmm1Δ strain but also those of all the single ERMES deletion mutant strains, including, unlike ChiMERA (Kornmann et al., 2009), mdm10Δ (Fig. 3 D), indicating that Mdm10 becomes dispensable when Mmm1's SMP domain is artificially targeted to the OMM. We tested whether Mdm12 and Mdm34 could equally rescue ERMES deficiency when artificially targeted to ER–mitochondria contact sites. To do so, we fused both proteins to both an N-terminal ER-targeting transmembrane domain (Sec66-TM) and a C-terminal mNG with the OMM-targeting Fis1-TM (Fig. 3 C). Mdm34, targeted to both mitochondria and the ER, could rescue the growth defects of mmm1Δ, mdm12Δ, and mdm10Δ strains, in addition to their cognate deletion strains, but to a lesser extent. Mdm12 failed to rescue the growth of even the mdm12Δ strain in the same setup (Fig. 3 D). Importantly, all artificially tethered ERMES fusion proteins were expressed and localized to foci between the ER and mitochondria, reminiscent of the wild-type ERMES complex (Figs. S4 and S5), and thus highlighting their correct folding and targeting.

Since Mmm1-mNG-Fis1 could rescue each ERMES-deficient strain individually, we next tested whether it could complement the combined quadruple ERMES deletion mutant

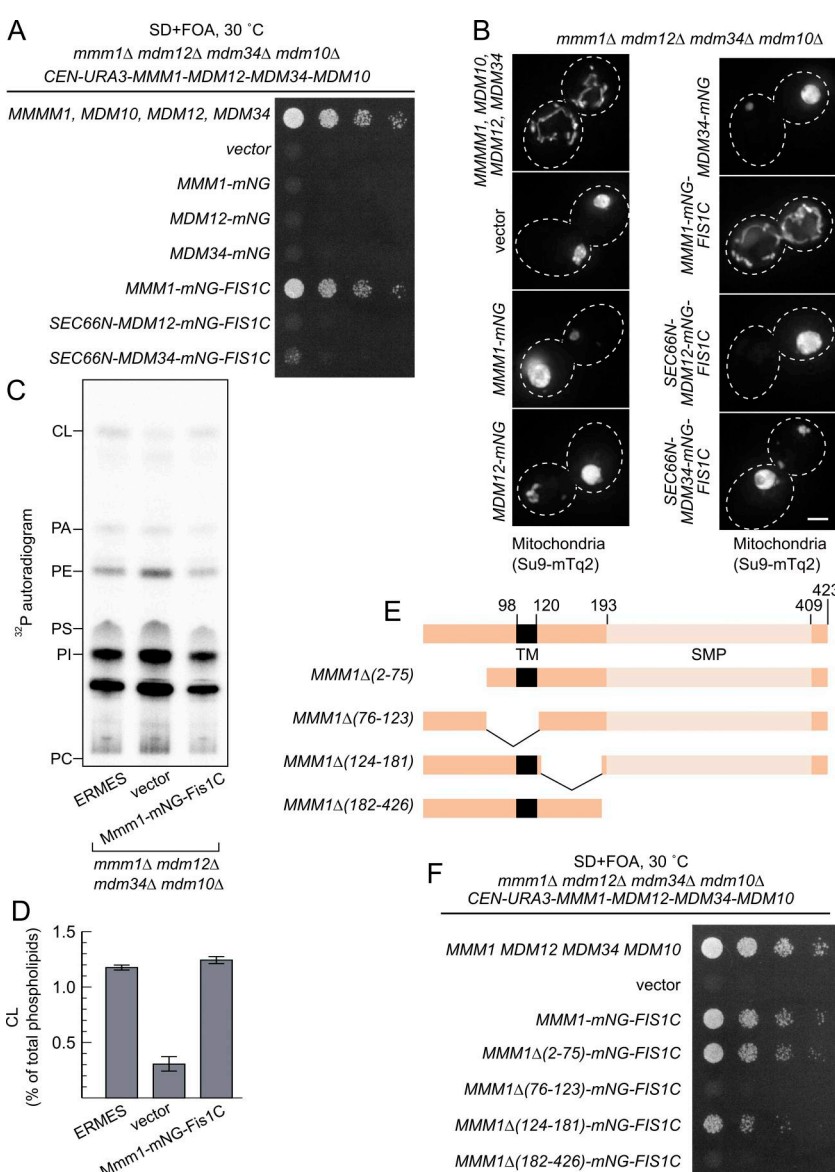

Figure 4. **SMP domain of Mmm1 is sufficient for rescue. (A)** Spot assay of ERMES quadruple deletion yeast with and without the expression of artificially tethered ERMES members. **(B)** Representative images of mitochondria from the yeast in A. Scale bars are 2 μm. **(C)** Representative thin layer chromatogram of total cell phospholipid extracts. PC, phosphatidylcholine; PI, phosphatidylinositol; PS, phosphatidylserine; PE, phosphatidylethanolamine; PA, phosphatidic acid; CL, cardiolipin. **(D)** Quantification of CL levels relative to total major phospholipids. Values are means ± SE (n = 3 independent experiments). **(E)** Schematic of Mmm1 truncation constructs used in F. **(F)** Spot assay of mmm1Δmdm12Δmdm34Δmdm10Δ yeast expressing truncations of Mmm1. Source data are available for this figure: SourceData F4.

(mmm1Δ/mdm12Δ/mdm34Δ/mdm10Δ). Indeed, Mmm1-mNG-Fis1C could functionally complement the quadruple ERMES deletion mutant (Fig. 4 A). In addition to the growth defect, Mmm1-mNG-Fis1C also rescued both the mitochondrial morphology phenotype and the reduced cardiolipin levels observed upon ERMES deletion (Fig. 4 B-D). Neither Mdm12 nor Mdm34 expressed in the same experimental setup rescued either growth or mitochondrial morphology (Fig. 4 A and B).

Mmm1 consists of an N-terminal ER-luminal domain dispensable for function (residues 1–97), an ER-inserted transmembrane domain (residues 98–120), and a cytosolic SMP domain (193–409), connected by linker sequences. To identify which of these domains are essential for the observed rescue activity, we generated a series of partial truncation and fusion constructs (Fig. 4 E). Deletion of the ER-luminal region (2–75) or the cytosolic linker region (124–181) did not affect the rescue activity, consistent with these domains being dispensable for Mmm1 function, yet deletion of the TM domain (76–123) or the

SMP domain (182–426) abolished the rescuing ability of the construct (Fig. 4 F). To assess whether Mmm1's TM domain played a special role besides targeting, we replaced it with Sec66-TM (Fig. 5 A) expressed from the ADH1 promoter. This substitution did not compromise the rescue activity (Fig. 5 B), indicating that the Mmm1 TM domain does not serve a special function besides ER attachment. To assess whether the order of the targeting sequences on the protein was important, we designed constructs where the N-terminal side of Mmm1's SMP domain was attached to an OMM-targeting domain (Tom70-TM) and its C terminus to an ER-targeting domain (Ubc6) (Fig. 5 A). Even in that inverted configuration, Mmm1's SMP domain could rescue the mutant growth defects (Fig. 5 B). By contrast, the SMP domain of Mdm12 or Mdm34 did not rescue the growth defect of the mutant yeast in either configuration (Fig. 5 B). Finally, to assess whether the lipid binding and transport of Mmm1's SMP domain was important for rescue, we used a molecular simulation-based algorithm (Grin et al., 2024; Maksymenko et al., 2023) to design specific mutations

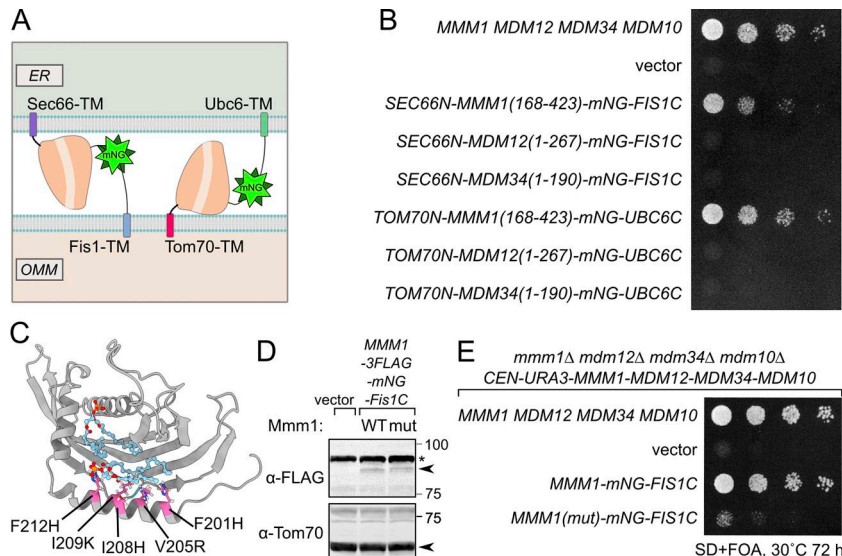

Figure 5. **Rescue of ERMES loss by Mmm1's SMP domain is independent of orientation and requires lipid transport activity. (A)** Schematic of constructs used in B. **(B)** Spot assay of *mmm1Δmdm12Δmdm34Δmdm10Δ* yeast strains expressing tethering constructs with different transmembrane and SMP domains. **(C)** Predicted mutant Mmm1 SMP domain structure based on Damietta and AlphaFold. The engineered mutations affecting lipid binding are indicated in pink. Phospholipids comes from the overlaid crystal structure of *Zygosaccharomyces rouxii* (PDB: 5YK6, [Jeong et al., 2017], RMSD = 0.806 Å). **(D)** Protein levels of Mmm1-mNG-Fis1C with or without mutations that impair lipid transfer activity. Total cell lysates were analyzed by immunoblotting with the indicated antibodies. Tom70 was used as loading controls. The asterisk indicates non specific signal. **(E)** Spot assay of *mmm1Δ mdm12Δ mdm34Δ mdm10Δ* yeast cells expressing *MMM1-mNG-Fis1C* constructs. The wild-type construct is compared to a lipid transport–deficient quintuple mutant of the Mmm1 SMP domain. Source data are available for this figure: SourceData F5.

predicted to prevent lipid transport by Mmm1 without affecting the overall fold and stability of the protein. The resulting quintuple mutant (Fig. 5 C) was used in the same mitochondrially targeted construct as its wild-type counterpart, and expressed as robustly (Fig. 5 D). Yet no rescue was observed despite robust localization at the ER–mitochondrial contacts (Fig. 5 E and Fig. S4, bottom panels). To conclude, the SMP domain of Mmm1 can function alone in lipid transfer between the ER and mitochondria, outside of the ERMES complex, and Mdm10 is likely responsible for solely tethering the complex, either full or partial to the mitochondrion.

**A minimal model for ERMES-mediated lipid transport**

The ERMES complex was originally discovered as an ER–mitochondria tether because its function could be substituted by an artificial ER–mitochondria tethering protein—ChiMERA (Kornmann et al., 2009). Subsequent structural and functional studies, however, established that ERMES is not merely a tether: it catalyzes interorganelle lipid transport through dedicated lipid transport domains in Mmm1, Mdm12, and Mdm34 (Kawano et al., 2018; Jeong et al., 2017; AhYoung et al., 2015; Ah Young et al., 2017; John Peter et al., 2022). This created a puzzling conundrum; how could an artificial tether without lipid transport activity rescue a compositionally rigid protein complex with demonstrated lipid transport activity? Our data rule out two simple explanations: redundancy between Mdm12 and Mdm34 in lipid transport and an involvement of Vps13 in ChiMERA-mediated rescue of ERMES deficiency. The latter was a particularly tempting model, since bolstering the Mcp1/Vps13 pathway (either by point mutations in Vps13 or by Mcp1 overexpression) represents the only other understood means to achieve ERMES rescue (Park et al., 2016; Lang et al., 2015; Kojima et al., 2016; Tan et al., 2013), as Vps13 itself was shown to mediate lipid transport (Li et al., 2020; Adlakha et al., 2022; Kumar et al., 2018).

Vps13 localizes to foci at ER–mitochondria contacts in a Mcp1 overexpression and Gem1-dependent fashion. Recently, ERMES foci have been shown to be sites of budding of MDCs—protrusions of OMM enriched in various OMM proteins that are important to survive amino acid stresses (Schuler et al., 2021). ER-mitochondrial Vps13 foci are most likely MDCs for three reasons: (1) They colocalized with Mdm34-mCherry–labelled ERMES foci, (2) they were more prominent and adopted a doughnut-like appearance when cells reached the saturation phase, and (3) their appearance was entirely dependent on Gem1. In fact, a microscopy-based screen identified Mcp1 as being enriched at MDCs (Hughes et al., 2016). Given the importance of lipid homeostasis on MDC formation (Xiao et al., 2024), it appears plausible that Vps13-mediated lipid transport plays a role in MDC biology.

We here argue that Mmm1 and Mdm10 might, by themselves, suffice to provide a core lipid exchange platform, when aided by additional tethering forces (model 3; Fig. 1 D). Strikingly, the SMP domain of Mmm1 alone, when artificially recruited to ER–mitochondria contact sites via N- and C-terminal targeting motifs, can rescue the complete loss of all ERMES components. This model 3 explains why ChiMERA expression fails to efficiently rescue loss of Mmm1 or Mdm10, in contrast to deletions of Mdm12 or Mdm34 and highlight a special role for Mmm1 and Mdm10 that cannot be substituted by tethering forces alone. Though in heterotetrameric ERMES complexes, Mdm12 and Mdm34 are still involved in efficient transport of lipids by organizing multiple SMP domains in tandem.

Whether wild-type *Saccharomyces cerevisiae* cells naturally contain an Mmm1–Mdm10 subcomplex remains unclear. If such forms exist, they are likely rare or transient, yet evidence from other species supports this two-protein model. In *Schizosaccharomyces pombe*, a genome-wide suppression screen revealed that Mmm1 overexpression could rescue lethality caused by defects in Mdm12, Mdm34, or both (Li et al., 2019). Importantly, this rescue required Mdm10, further emphasizing the existence of functional subcomplexes outside of the canonical ERMES. In contrast, in budding yeast we find that Mmm1 overexpression does not detectably ameliorate the phenotypes of

single *mdm12* and *mdm12/vps13* mutants (Fig. S3 B and C). This species specificity suggests divergent evolutionary paths: while an Mmm1–Mdm10 subcomplex may retain autonomous function in fission yeast, it has likely lost this capacity in budding yeast in the absence of external tethering forces.

In summary, our findings provide a resolution to the paradox that a purely artificial tether can functionally compensate for the absence of LTPs.

## Materials and methods

### Plasmids and yeast strains
Yeast gene deletions were achieved by homologous recombination of selective markers including 45 base homology arms to upstream and downstream of the open reading frame (Longtine et al., 1998). Overexpression of the endogenous *MCP1, VPS39*, and *MMM1* genes was achieved by integrating a GPD promoter cassette (Janke et al., 2004). Marker-free gene deletions were achieved by the CRISPR/Cas9 system (Soreanu et al., 2018).

The quintuple point mutant of Mmm1 (F201H/V205R/I208H/I209K/F212H)) that prevented lipid transport was designed using Damietta to ensure protein folding and stability (Grin et al., 2024; Maksymenko et al., 2023). The residues chosen to mutate were informed by those coordinating lipids in crystal structures (PDB: 5YK6).

### Media
Yeast cells were cultured in SD media (0.67% yeast nitrogen base without amino acids, 2% glucose) supplemented with 20 µg ml$^{-1}$ each of adenine sulfate, L-histidine-HCl, L-tryptophan, and uracil and 30 µg ml$^{-1}$ each of L-leucine and L-lysine-HCl. For spot assay, 1 mg ml$^{-1}$ 5-fluoroorotic acid was added. For microscopic observation, the concentration of adenine sulfate was doubled.

### Spot assay (plasmid shuffling)
Yeast strains were grown to saturation in SD lacking tryptophan and uracil, then diluted 100-fold with SD lacking tryptophan and cultured for 24 h. Then, the cultures were serially diluted five-fold, spotted onto agar plates containing 5-fluoroorotic acid, and incubated at 30°C for 48 h.

### Tetrad analysis
Diploid yeast strains were sporulated by patching cells onto agar plates containing 1% yeast extract, 2% peptone, and 1% potassium acetate. After the appearance of tetrads (5–7 days), they were dissected onto YPD and subsequently genotyped by replica plating onto media to select for the markers of interest.

### Fluorescence microscopy
For Fig. 2, saturated cultures of yeast were reseeded in appropriate selective media at a very low density so that they would still be in log phase after at least 16 h of growth. Then, ~500,000 cells were plated on a microscope slide with a coverslip on top. Images were obtained using an UltraView IX81 Olympus spinning disc confocal microscope equipped with a 100× oil immersion objective lens (NA = 1.4). Image acquisition, using Volocity software, was carried out at room temperature. For Fig. 4, yeast strains were grown to log phase and placed on a microscope slide with a coverslip. Cells were observed at room temperature using a DeltaVision Elite system (Cytiva) equipped with a 100× objective lens (UPLSAPO, NA/1.40; Olympus) and an sCMOS camera (Edge5.5; PCO). Z-sections were captured every 0.2 or 0.4 µm from the top to the bottom surface of yeast cells. Deconvolution was performed using SoftWoRx software (Cytiva), and the acquired images were processed and analyzed with Fiji software. For Fig. S1 D, images were acquired using a DeltaVision MPX microscope (Applied Precision) equipped with a 100× 1.40 NA oil UplanS-Apo objective lens (Olympus), a multicolor illumination light source, and a Cool-SNAPHQ2 camera (Roper Scientific). Image acquisition was done at room temperature. Images were deconvolved with SoftWoRx software using the manufacturer's parameters.

### Image analysis and statistics
To ascertain whether Vps13^GFP puncta preferentially localize at ERMES foci, first, cells were scored for whether they had colocalization of Vps13^GFP and Mdm34-mCherry signal. This was determined to be the case if at least half of the Mdm34-mCherry pixels overlapped with the Vps13^GFP puncta signal. The percentage of cells with colocalization for each repeat was then compared with the expected probability that a Vps13^GFP puncta would randomly overlap with Mdm34-mCherry. This expected value was calculated by quantifying the size of the thresholded Vps13^GFP puncta and dividing it by the mitochondrial mass (calculated from the thresholded signal of the mitochondrial marker mtBFP), i.e., the chance that a Vps13^GFP puncta would fall at any position of the mitochondria. The percentage of cells showing colocalization (observed) was then compared with this expected value using a chi-squared test calculated in GraphPad Prism.

### Phospholipid analysis
Yeast cells were grown at 30°C to saturation in SD media. The cultures were diluted 100-fold with SD containing 2 µCi/ml of [$^{32}$P]phosphate (NEX053; Revvity) and further incubated at 30°C for 24–48 h to reach late-log phase. After incubation, 1 ml of cells was harvested, resuspended in 100 µl of ethanol-water (4:1, by volume), and then heated at 95°C for 15 min to extract lipids (Hanson and Lester, 1980). Cell debris was removed by centrifugation (20,000 × g, 1 min). The extracted lipids were dried under a stream of nitrogen gas and purified by the Folch method (Folch et al., 1957). The lipids were applied to a thin-layer chromatography plate with concentrating zone (1.11845.0001; Merck), which had been impregnated with 18 g/L boric acid in ethanol, and separated with chloroform/ethanol/water/triethylamine (30:35:7:35, by volume) (Vaden et al., 2005). $^{32}$P-labeled phospholipids were detected and quantified by autoradiography with a phosphor imaging plate (BAS IP MS 2025 E; Cytiva) and a Typhoon FLA-7000 imaging analyzer (Cytiva).

### Analysis of total cellular proteins
Yeast cells were grown at 30°C to log phase. Cells were collected and incubated in ice-cold 0.1 M NaOH for 10 min (Kushnirov,

2000). NaOH-treated cells were sedimented, resuspended in SDS sample buffer, and then heated at 95°C for 5 min to extract proteins. Proteins were separated by SDS-PAGE and transferred to PVDF membranes (Immobilon-FL, Merck). The membranes were blocked with 5% skim milk in TBS (25 mM Tris-HCl [pH 7.4]) 137 mM NaCl, and 2.68 mM KCl) for 1 h. Membranes were incubated with primary antibodies in 5% skim milk in TBS containing 0.05% Tween 20 (TBST) overnight at 4°C and washed three times with TBST (5 min for each wash). Membranes were then incubated with secondary antibodies in TBST containing 0.1 g/L SDS for 1 h and again washed three times with TBST. Signals were detected with an Odyssey DLx scanner (LI-COR).

### Structural predictions

Structural predictions of the Mmm1–Mdm10 dimer were made using AlphaFold 3 (Abramson et al., 2024) and visualized using ChimeraX (Pettersen et al., 2021).

### Online supplemental material

Fig. S1 shows Vps13 forms puncta on mitochondria. Fig. S2 shows effect of Gem1 deletion on Vps13 localization. Fig. S3 shows tetrad analysis of ChiMERA in ERMES deletion yeast. Fig. S4 shows subcellular localization of ERMES fusion proteins. Fig. S5 shows protein levels of the artificially tethered proteins used in this study. Source data shows uncropped gels and thin layer chromatography autoradiograph for Figs. 4, 5, and S5. Table S1 show all plasmids used in this study. Table S2 shows yeast strains used in this study. Table S3 shows antibodies used in this study.

## Data availability

All data are available in the published article and its online supplemental material.

## Acknowledgments

The authors gratefully acknowledge the Micron Advanced Bioimaging Facility (supported by Wellcome Strategic Awards 091911/B/10/Z and 107457/Z/15/Z) for their support and assistance in this work.

This work was funded by Wellcome Trust grant 214291/Z/18/Z and EPA Cephalosporin Fund grant CF 412 (awarded to B. Kornmann); Japan Society for the Promotion of Science (JSPS) KAKENHI 15H05705, 2222703, 20H04912, 20H05689, 20H05929, 25H00978, and 25H21752 (to T. Endo); JSPS KAKENHI 17K18230 and 25840020 (to S. Kawano); Japan Science and Technology Agency Core Research for Evolutional Science and Technology (CREST) grant JPMJCR12M1 (to T. Endo); Japan Agency for Medical Research and Development CREST grant 21gm1410002h0002 (to T. Endo); and a grant from Takeda Science Foundation (to T. Endo). T. Hirashima (23KJ2067) was supported by a Research Fellowship for Young Scientists from the Japan Society of the Promotion of Science.

Author contributions: Christian Covill-Cooke: conceptualization, formal analysis, investigation, methodology, supervision, visualization, and writing—original draft, review, and editing. Takashi Hirashima: conceptualization, funding acquisition, investigation, methodology, resources, visualization, and writing—original draft, review, and editing. Shin Kawano: investigation. Joe Ganellin: formal analysis, investigation, and writing—review and editing. Andrew Moody: investigation. Sabine N.S. van Schie: conceptualization, supervision, and writing—review and editing. Arun T. John Peter: investigation. Chika Horie Saito: investigation and resources. Toshiya Endo: conceptualization, funding acquisition, project administration, supervision, visualization, and writing—original draft, review, and editing. Benoît Kornmann: conceptualization, data curation, funding acquisition, methodology, project administration, resources, supervision, visualization, and writing—original draft, review, and editing.

Disclosures: The authors declare no competing interests exist.

Submitted: 26 November 2024

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

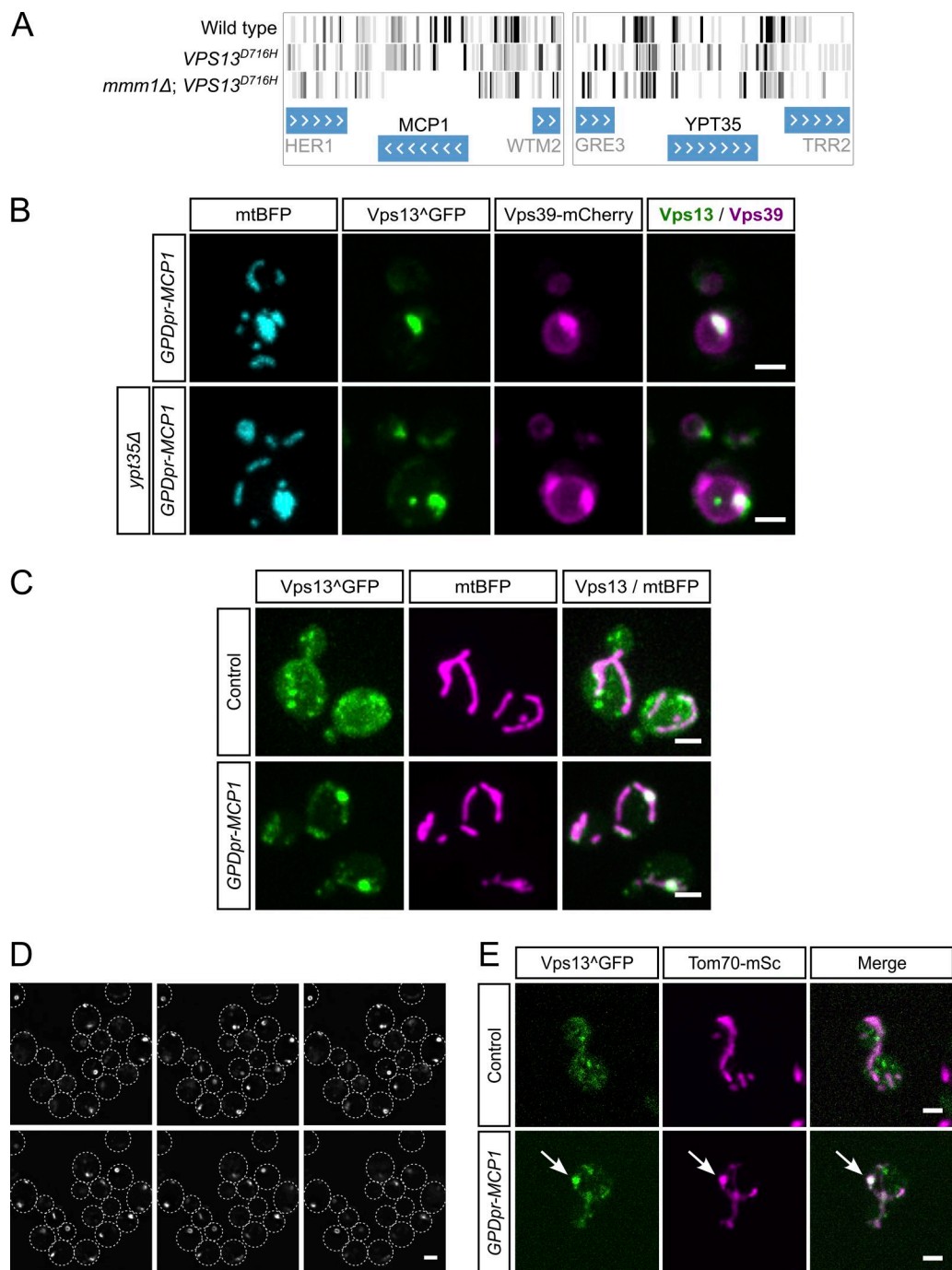

**Figure S1.** **Vps13 forms puncta on mitochondria. (A)** Comparison of transposon density maps for *MCP1* and *YPT35* in wild-type, *VPS13^D716H^*, and *mmm1Δ VPS13^D716H^* yeast using saturated transposon analysis in yeast (SATAY) (Michel et al., 2017). **(B)** Representative images of Vps13^GFP localization to vCLAMPs in Vps39 and Mcp1 co-overexpression conditions, both with and without Ypt35. mtBFP is a mitochondrial marker. **(C)** Representative images of Vps13^GFP both with and without the overexpression of *MCP1*. **(D)** Individual Z planes of Vps13^GFP when cells are in stationary phase. **(E)** Representative images of Vps13^GFP and Tom70-mScarlet in control and Mcp1-overexpressing yeast. Scale bars are 2 μm. vCLAMPs, vacuolar and mitochondrial patches.

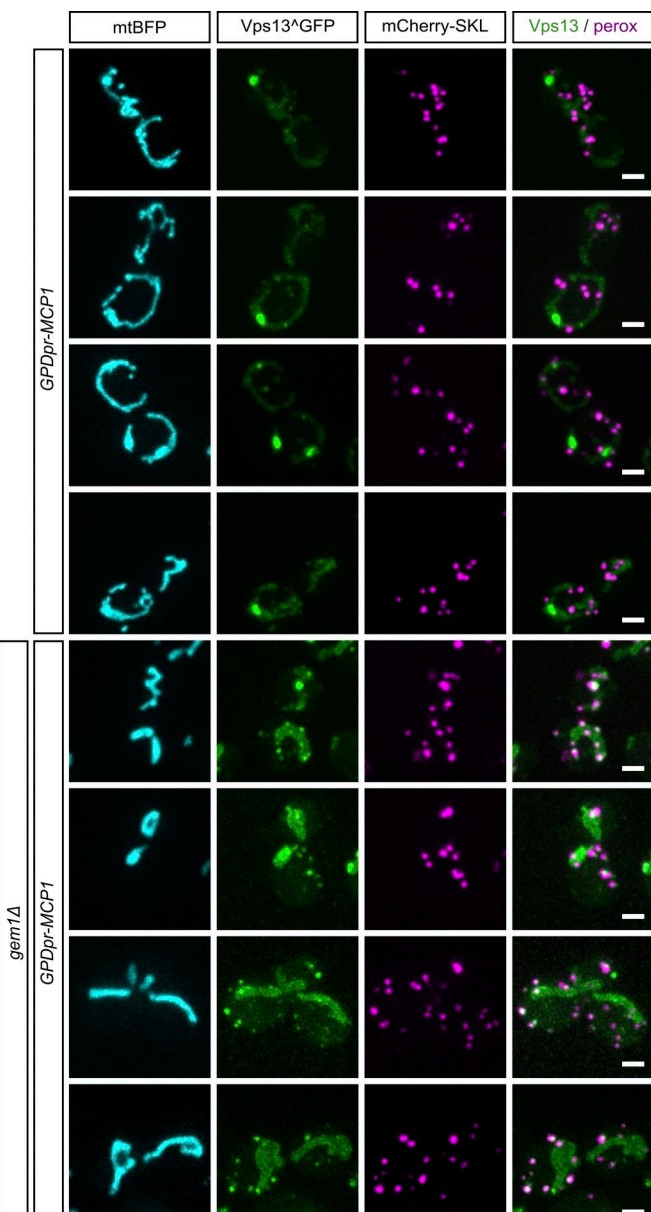

Figure S2. **Effect of Gem1 deletion on Vps13 localization.** Multiple representative images of Vps13^GFP subcellular localization in comparison with mitochondria (mtBFP) and peroxisomes (mCherry-SKL) in Mcp1 overexpression conditions, both with and without Gem1. Scale bars are 2 µm.

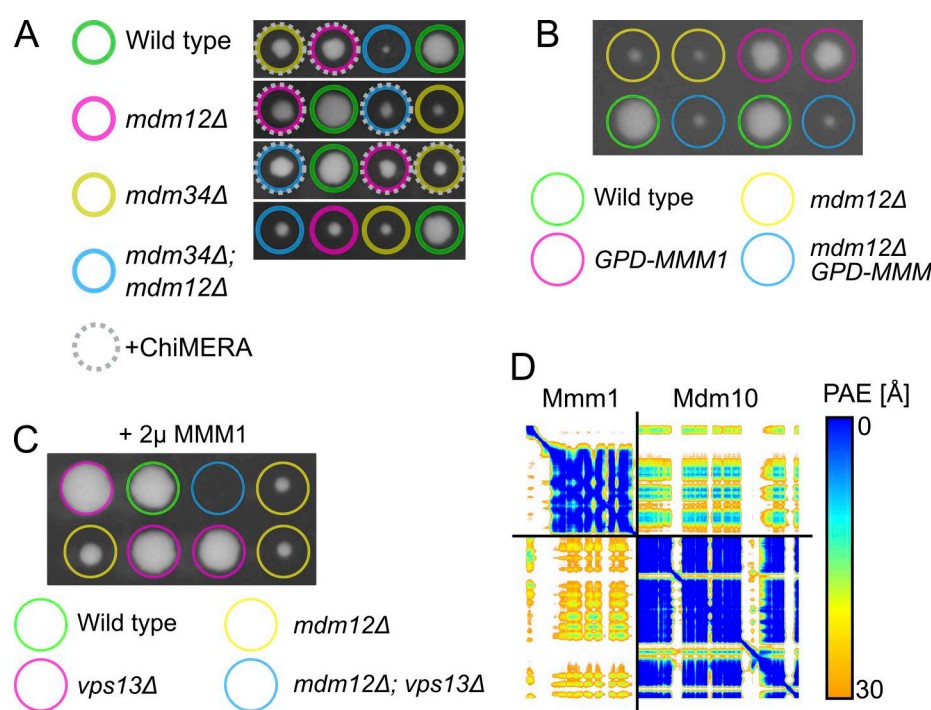

Figure S3. **Tetrad analysis of ChiMERA in ERMES deletion yeast. (A)** Representative tetrads from the sporulation of *MDM12/mdm12Δ MDM34/mdm34Δ* diploids expressing ChiMERA. **(B)** Tetrad analysis of the effect of Mmm1 overexpression (with an integrated GPD promoter on the endogenous gene) on growth of *mdm12Δ*. **(C)** Tetrad analysis of the effect of Mmm1 overexpression (with GPD promoter on a 2μ plasmid) on growth of *mdm12Δ/vps13Δ*. **(D)** Predicted aligned error plot for the AlphaFold model in Fig. 3 B.

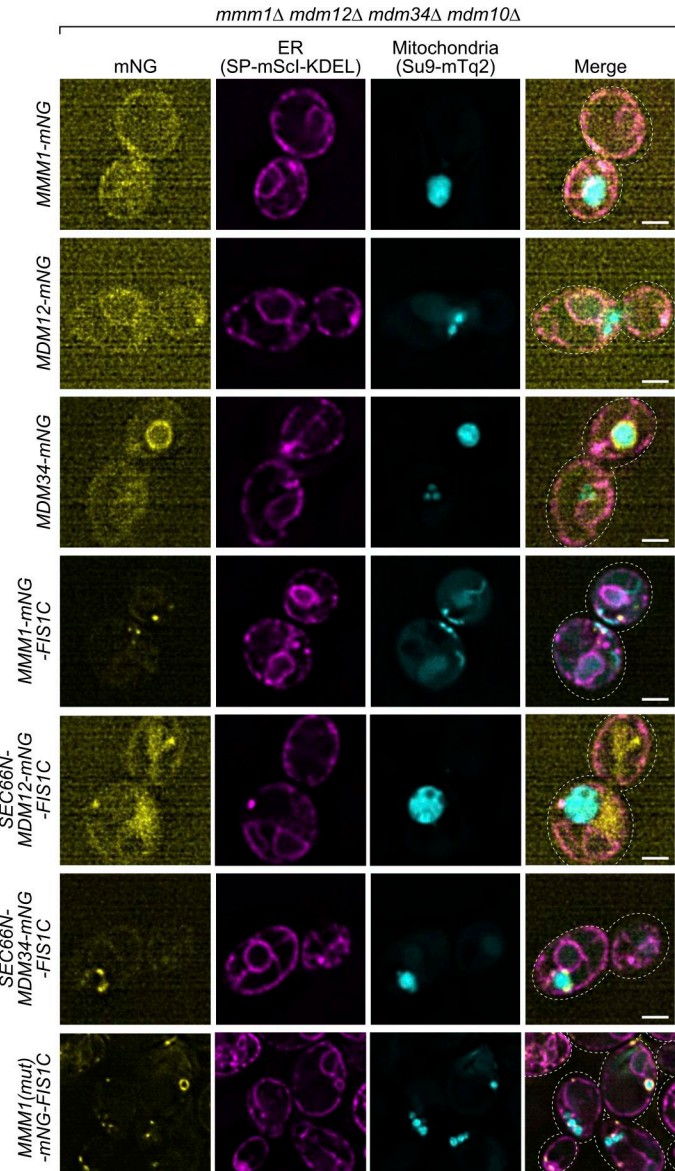

Figure S4. **Subcellular localization of ERMES fusion proteins.** Representative images of localization of mNG-fused proteins. Kar2(1–45)-mScarletI-KDEL (SP-mScI-KDEL) and Su9(1–69)-mTurquoise2 (Su9-mTq2) are ER and mitochondrial markers, respectively. Scale bars are 2 µm.

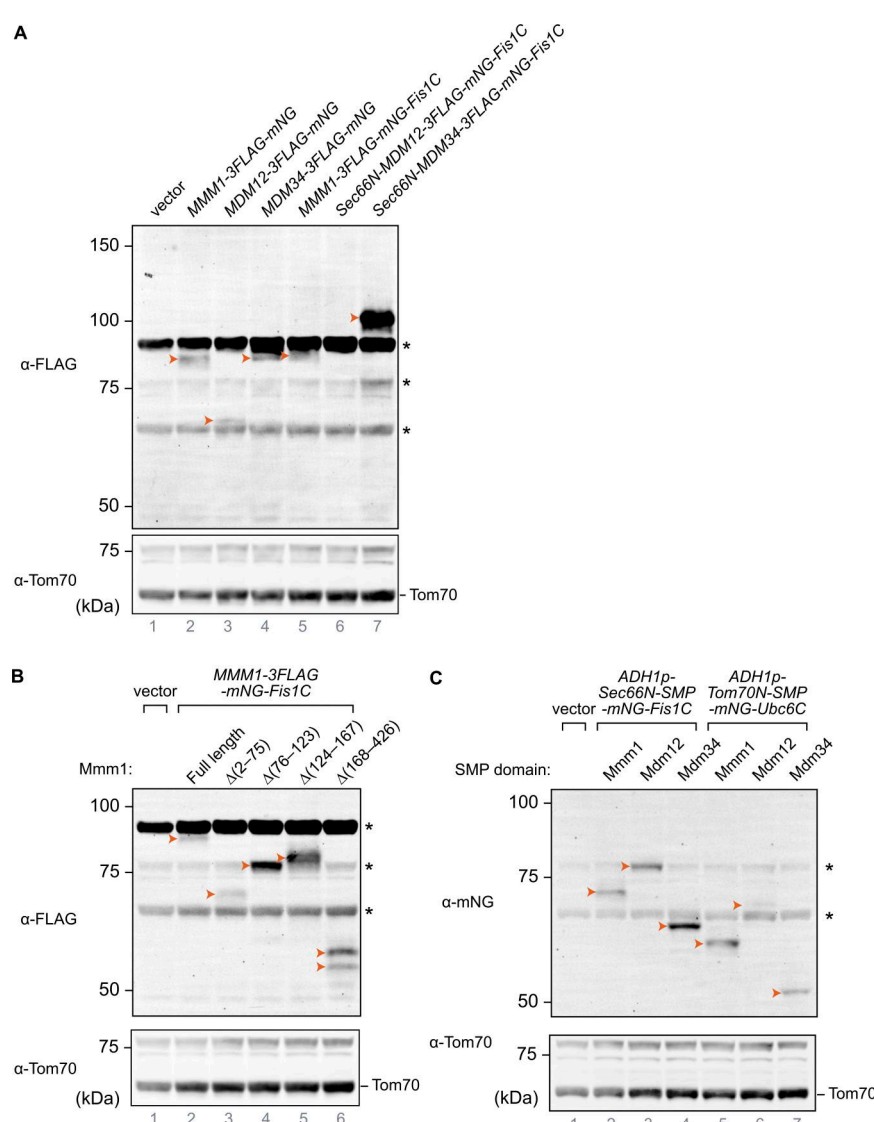

Figure S5. **Protein levels of the artificially tethered proteins used in this study. (A–C)** Total cell lysates were analyzed by immunoblotting. Arrowheads indicate the expressed proteins. Asterisks indicate nonspecific signals. Tom70 was used as loading controls. **(A)** Protein levels of artificially tethered ERMES members. **(B)** Protein levels of Mmm1-mNG-Fis1C with partially truncated Mmm1. **(C)** Protein levels of tethering constructs with different transmembrane and SMP domains. Source data are available for this figure: SourceData FS5.

Provided online are Table S1, Table S2, and Table S3. Table S1 shows all plasmids used in this study. Table S2 shows yeast strains used in this study. Table S3 shows antibodies used in this study.

