## [Peer Review File · The Journal of Cell Biology]

Mitochondrially-tethered Mmm1 can function as sole lipid transporter at ER-mitochondria contacts

Christian Covill-Cooke, Takashi Hirashima, Shin Kawano, Joe Ganellin, Andrew Moody, Sabine van Schie, Arun John Peter, Chika Saito, Toshiya Endo, and Benoit Kornmann

Corresponding Author(s): Benoit Kornmann, University of Oxford

Review Timeline:

Submission Date:	2024-11-26
Editorial Decision:	2025-01-16
Revision Received:	2025-09-23
Editorial Decision:	2025-10-22
Revision Received:	2026-03-03
Editorial Decision:	2026-03-26
Revision Received:	2026-04-07

Monitoring Editor: Thomas Langer

Scientific Editor: Dan Simon

Transaction Report:

DOI: <https://doi.org/10.1083/jcb.202411196>

January 16, 2025

Re: JCB manuscript #202411196

Benoit Kornmann
University of Oxford

Dear Dr. Kornmann,

Thank you for submitting your manuscript entitled "Compositional Flexibility of the ER-Mitochondria Encounter Structure." Your manuscript has been assessed by expert reviewers, whose comments are appended below. Although the reviewers express potential interest in this work, significant concerns unfortunately preclude publication of the current version of the manuscript in JCB.

You will see that Reviewers #2&3 point out that currently there is no evidence for the existence of different ERMES subcomplexes in wildtype cells. We agree that this is a crucial issue that would need to be convincingly addressed with new data in order for us to consider this study further. Functional studies of ERMES subcomplexes would not be expected for a Report but if you do have such data then the manuscript could be expanded into a full Article. Reviewers #1&2 also ask for additional controls and quantifications to better support the claims that Vps13 foci are mitochondria-derived compartments and that ChiMERA can rescue the growth defect of *mdm34Δ/mdm12Δ* strains, as well as text and figure changes to clarify the roles of Mdm12 and Mdm34 and better explain the models that are being tested. These are also important points that need to be addressed.

Please let us know if you are able to address the major issues outlined above and wish to submit a revised manuscript to JCB. Note that a substantial amount of additional experimental data likely would be needed to satisfactorily address the concerns of the reviewers. The typical timeframe for revisions is three to four months. If you anticipate any difficulties in meeting this aforementioned revision time limit, please contact us and we can work with you to find an appropriate time frame for resubmission. Please note that papers are generally considered through only one revision cycle, so any revised manuscript will likely be either accepted or rejected.

If you choose to revise and resubmit your manuscript, please also attend to the following editorial points. Please direct any editorial questions to the journal office.

GENERAL GUIDELINES:

Text limits: Character count for a Report is < 20,000; a full Research Article is < 40,000, not including spaces. Count includes title page, abstract, introduction, the joint Results & Discussion, and acknowledgments. Count does not include materials and methods, figure legends, references, tables, or supplemental legends.

Figures: A Report may include up to 5 main text figures; a full Research Article may have up to 10 main text figures. To avoid delays in production, figures must be prepared according to the policies outlined in our Instructions to Authors, under Data Presentation, <https://jcb.rupress.org/site/misc/ifora.xhtml>. All figures in accepted manuscripts will be screened prior to publication.

Supplemental information: There are strict limits on the allowable amount of supplemental data. Reports may have up to 3 supplemental figures; a full Research Article may have up to 5 supplemental figures. Up to 10 supplemental videos or flash animations are allowed. A summary of all supplemental material should appear at the end of the Materials and methods section.

Please note that JCB now requires authors to submit Source Data used to generate figures containing gels and Western blots with all revised manuscripts. This Source Data consists of fully uncropped and unprocessed images for each gel/blot displayed in the main and supplemental figures. Since your paper includes cropped gel and/or blot images, please be sure to provide one Source Data file for each figure that contains gels and/or blots along with your revised manuscript files. File names for Source Data figures should be alphanumeric without any spaces or special characters (i.e., SourceDataF#, where F# refers to the associated main figure number or SourceDataFS# for those associated with Supplementary figures). The lanes of the gels/blots should be labeled as they are in the associated figure, the place where cropping was applied should be marked (with a box), and molecular weight/size standards should be labeled wherever possible. Source Data files will be made available to reviewers during evaluation of revised manuscripts and, if your paper is eventually published in JCB, the files will be directly linked to specific figures in the published article.

If you choose to resubmit, please include a cover letter addressing the reviewers' comments point by point. Please also highlight all changes in the text of the manuscript.

Regardless of how you choose to proceed, we hope that the comments below will prove constructive as your work progresses. We would be happy to discuss them further once you've had a chance to consider the points raised. You can contact the journal office with any questions at cellbio@rockefeller.edu.

Thank you for thinking of JCB as an appropriate place to publish your work.

Sincerely,

Thomas Langer, PhD
Monitoring Editor
Journal of Cell Biology

Dan Simon, PhD
Scientific Editor
Journal of Cell Biology

Reviewer #1 (Comments to the Authors (Required)):

The ERMES complex is the most important lipid transporter between the ER and mitochondria in yeast. However, its exact mechanism and the roles of its individual subunits are not well understood. In their present manuscript Covill-Cooke et al. proposed and tested several models and found that lipid transfer activity of Mmm1 alone is sufficient when both organellar membranes are closely tethered. The authors report interesting observations and I found the rescue of the *mmm1 mdm12 mdm34 mdm10* quadruple mutant by *MMM1-mNG-FIS1C* particularly striking. However, some of their statements in the text should be supported by more solid and/or quantitative data including all relevant controls, as outlined below.

Specific points

1. The authors suggest several scenarios for rescue of lipid transport by ChiMERA, which are depicted in Figure 1. I find the third scenario a bit confusing. The text says that Mmm1 might serve as the sole lipid transport protein (line 80), however, in Fig. 1D Mmm1 is shown in a complex with Mdm10. Also, the legend suggests that Mmm1 and Mdm10 provide tethering force? I understand that the tethering force should be provided by ChiMERA. In lines 148-151 the authors describe that Mdm10 is devoid of a lipid transport domain and propose that it serves as a recruitment factor for Mmm1. If this is the case, then Figs. 1D and 3A are misleading as the cartoon suggests that lipids are shuttled through Mdm10 as well.
2. The experiment shown in Fig. 2B and C should show a sample without *Mcp1* overexpression analyzed in parallel. What does "+ve" mean? How many cells were analyzed in how many independent experiments?
3. The authors claim that *Vps13* foci formed upon *Mcp1* overexpression represent mitochondria-derived compartments. Unfortunately, they don't present any direct experimental evidence supporting this statement. Also, I'm unable to see in Fig. S1D that *Vps13* foci form in proximity of ERMES, which was not labelled in this experiment. Moreover, the authors say that *Gem1* deletion abrogated the punctate localization of *Vps13*. Unfortunately, they show only a single cell. More quantitative data are required here. Furthermore, Fig. S1D should include outlines of the cells that were imaged.
4. The authors claim that ChiMERA rescues the growth defect of a *mdm34 mdm12* double mutant. This was tested by tetrad dissection. The authors show only two colonies with and two without the rescuing plasmid. More quantitative data are required to support this statement. The authors should perform drop dilution tests or obtain growth curves of at least three biological replicates.
5. The authors compare rescue of *mdm10 delta* by *MMM1-mNG-FIS1C* with rescue by ChiMERA (line 157). The lack of rescue by ChiMERA should be shown in the same experiment in Fig. 3D.
6. Do the mNG constructs used in Figs. 3 and 4 form punctate structures similar to native ERMES complexes? This could be easily tested by fluorescence microscopy.

Reviewer #2 (Comments to the Authors (Required)):

The authors investigate how an artificial tether between ER and mitochondria (ChiMERA) without lipid transfer activity can rescue defects in the ER-mitochondria tether and lipid transfer ERMES complex. The authors present a systematic analysis of how different remaining ERMES subunits might work together to provide some residual function when ER and mitochondria are

synthetically tethered. Using genetic approaches, the authors nicely demonstrate that Mmm1 provides sufficient remaining PL transfer activity to rescue the ERMES defect associated growth phenotype in artificial contexts of ChiMERA expression or when itself being synthetically modified to increase membrane tethering.

This is a neat and focused study, but with likely marginal implications for a broader readership. Clearly, the proposed Mmm1-Mdm10 complex does not provide sufficient PL transfer activity under physiological conditions to promote mitochondria morphology or function without artificially stabilized ER-mitochondria tethering. However, differential residual activity in multicomponent complexes is not a new phenomenon and the actual biological implications in the wildtype setting remain unclear. The authors imagine the existence of ERMES complexes with differential subunit composition to for example bridge different distances of MCS, but do not provide any evidence for their actual existence in cells. Thus, the title of the manuscript implies cell biology not really shown in the manuscript.

To make the manuscript more appealing to a broader readership it would be important to (1) provide evidence for the existence and function of the proposed subcomplexes in wildtype cells or (2) substantiate the findings with an evolutionary link to other organisms with minimal ERMES complexes. Otherwise, the manuscript appears to address a rather technical or niche problem of unclear physiological relevance.

Reviewer #3 (Comments to the Authors (Required)):

The ER-mitochondria encounter structure ERMES consists of four obligatory components, deletion of each resulting in a severe phenotype. ERMES is thought to assemble between the ER and mitochondrial membranes to channel lipids between the two organelles. An artificial tether without capacity to transfer lipids can rescue the deletion of the ERMES component Mdm12 or Mdm34, but deletions of Mdm10 or Mmm1 are only rescued to a low extent. This has been mysterious because Mdm12 and Mdm34 consist of similar lipid transfer domains as Mmm1.

This manuscript tests three possible models that could explain this mystery. Two models are ruled out experimentally, while the third one turns out to be the likely explanation. The authors conclude that of the four ERMES components, a minimal assembly of Mmm1 and Mdm10 is sufficient for function, provided that ER and mitochondria are tethered otherwise, such as through the artificial tether ChiMERA.

This study consists of a set of elegant, technically demanding and well carried-out experiments that shed new light on a long-standing question and is thus of wide interest to the cell biology community. I recommend it to be published, but have some suggestions for revisions as described below.

Major:

Overall, the conclusions of this paper are well supported by the results. However, I find that the title, abstract and the introduction are somewhat misleading, as they imply that ERMES assemblies of variable composition do exist. While I agree with the authors that their results raise that possibility, it remains a hypothesis. There is no evidence that in wild type cell (i.e. in absence of ChiMERA expression) the Mmm1-Mdm10 complex makes up a significant fraction of lipid trajectories between the ER and mitochondria, or even exists.

In addition, the organisation of ERMES into the sequence Mmm1-Mdm12-Mdm34-Mdm10 is derived from biochemistry by Ellenrieder et al (Nat Comm 2021), not from cryotomography, which rather used the biochemistry results to interpret a low-resolution average structure. The statement in lines 44-49 of the introduction is thus a bit misleading, and Ellenrieder et al should be cited.

Furthermore, the manuscript does not fully resolve the mystery of ChiMERA rescue, because the roles of Mdm12 and Mdm34 remain unaddressed. These two components do likely active lipid transfer domains, and their deletion growth phenotype is not fully rescued by ChiMERA to wild type level (Figure 2G in this manuscript and Kornmann et al 2009, Fig 2A), albeit the rescue is more pronounced than for Mdm10 and Mmm1. Thus, they must have a function beyond only tethering.

Minor:

Lines 138-139 (and other instances) I do not think the term 'redundant' is used correctly regarding Mdm12 and Mdm34. The authors mean to say that Mdm12 and Mdm34 cannot replace each other (and I agree based on their results), but redundant can also mean dispensable; thus not needed or superfluous. In that sense, saying that Mdm12 and Mdm34 are not redundant seems misleading with regard to the results. The authors might want to reconsider the use of the term redundant.

Lines 90-97: It is unclear whether the authors mean to say that these Vps13 spots appear near ERMES in wild type cells or when Mcp1 is overexpressed.

Figure 1 is perhaps not necessary. The key panels are shown again in Figures 2 and 3.

Line 143: "providing tethering force" The authors might want to reconsider this phrasing; there is no indication that Mdm12 or Mdm34 are able to provide any force.

Line 148: "...gap induced by" It is not known what gap ChiMERA mediated tethering induces; what the authors mean is the predicted gap. It has not been experimentally determined.

Line 117: Typo, components should be component

Line 471 : Typo, mdm34-mCherry should be Mdm34-mCherry

Dear editor, dear Thomas,

Thank you very much for handling our manuscript.

Following the chat that Toshi has had with you in Japan, we present here our revised manuscript as well as our responses to the reviewer's comments.

The reviewers were generally positive about our work, finding it a "neat and focused study" and "a set of elegant, technically demanding and well carried-out experiments that shed new light on a long-standing question". However, three main criticisms were raised. The first one is a set of technical concerns. We have taken these on board to present a stronger and better-controlled manuscript. The second is that, while our data hint at the existence of a functional Mmm1-Mdm10 subcomplex, we did not show that this complex exists, let alone functions, outside of the conditions using the artificial ChiMERA. This is a fair criticism, and we agree that parts of our original manuscript, including its title, were referring to speculations rather than established facts. We have corrected this and refocused on the study's main findings (amended text is highlighted in yellow in the revised manuscript). We would like, however, to draw reviewers' attention to the evidence from *S. pombe*, where a similar Mmm1-Mdm10 subcomplex appears to be functional even in the absence of artificial tethering. We therefore consider it legitimate to speculate in the discussion section on the existence and function of such a subcomplex. The third main criticism (related to the second one) is that in the absence of solid evidence for the existence of a subcomplex in normal cells, our findings may be of niche interest. We beg to differ. The paradox raised by the fact that the ChiMERA construct, a pure tether without lipid transport activity, could functionally replace the function of lipid transporters has been a source of controversy in the field (Nguyen et al. 2012, Voss et al. 2012). Only after the structural characterization of ERMES components and related proteins, as well as the identification of a partially redundant pathway (Mcp1/Vps13), did the idea of ERMES-mediated interorganelle lipid transport become widely accepted. However, this still does not resolve the original paradox. The present study can be seen as the last nail in the coffin of this controversy, as we can now rationalise all previous observations.

Please see below for our responses to the reviewer's points

Reviewer #1 (Comments to the Authors (Required)):

The ERMES complex is the most important lipid transporter between the ER and mitochondria in yeast. However, its exact mechanism and the roles of its individual subunits are not well understood. In their present manuscript Covill-Cooke et al. proposed and tested several models and found that lipid transfer activity of

Mmm1 alone is sufficient when both organellar membranes are closely tethered. The authors report interesting observations and I found the rescue of the mmm1 mdm12 mdm34 mdm10 quadruple mutant by MMM1-mNG-FIS1C particularly striking. However, some of their statements in the text should be supported by more solid and/or quantitative data including all relevant controls, as outlined below.

Specific points

1. The authors suggest several scenarios for rescue of lipid transport by ChiMERA, which are depicted in Figure 1. I find the third scenario a bit confusing. The text says that Mmm1 might serve as the sole lipid transport protein (line 80), however, in Fig. 1D Mmm1 is shown in a complex with Mdm10. Also, the legend suggests that Mmm1 and Mdm10 provide tethering force? I understand that the tethering force should be provided by ChiMERA. In lines 148-151 the authors describe that Mdm10 is devoid of a lipid transport domain and propose that it serves as a recruitment factor for Mmm1. If this is the case, then Figs. 1D and 3A are misleading as the cartoon suggests that lipids are shuttled through Mdm10 as well.

We would like to thank the reviewer for pointing that out. We agree with the reviewer that the depiction of Mdm10 in the original figures is somewhat misleading. We have therefore revised the figures to clarify that Mdm10 is not a lipid-transport protein

Regarding the role of Mdm10, ChiMERA brings the two membranes into close proximity, which is crucial for Mmm1 to perform its lipid-transfer function in place of the full ERMES complex. With this short distance between the organelles, Mmm1 still requires an anchor to physically link to the mitochondrial membrane; otherwise, lipid transfer into the membrane cannot occur efficiently. Mdm10 appears to serve this anchoring role, as AlphaFold3 predicts a possibility of formation of a complex between Mdm10 and Mmm1. This complex is likely very unstable and therefore cannot provide tethering force by itself. We have revised the legend to Fig. 1D to reflect this point.

2. The experiment shown in Fig. 2B and C should show a sample without Mcp1 overexpression analyzed in parallel. What does "+ve" mean? How many cells were analyzed in how many independent experiments?

We apologize for the lack of clarity of our presentation ("+ve" was meant to mean "positive"). The data in Figure 2B-C are an analysis of the relationship between Vps13 foci and ERMES. A comparison with non-Mcp1-overexpressing cells was not included because mitochondrial Vps13 foci do not occur without Mcp1 overexpression. We added an example image of Vps13 in the absence of Mcp1 overexpression in Supp Fg 1b. However, as Figure 2C analyzes the probability of these Vps13 foci

randomly colocalizing with Mdm34-mCh signal, non-Mcp1 overexpressing cells cannot be included in this analysis. We have statistically compared the observed and expected results by Chi-squared analysis. As this analysis is clearly described in the Material and Methods, we have now added “(see Materials and Methods)” to the main text. We have also added extra details to the legend of Figure 2. The axis labels have been changed from “+ve” to “positive”, and the number of cells and replicates have been added to the legend.

3. The authors claim that Vps13 foci formed upon Mcp1 overexpression represent mitochondria-derived compartments. Unfortunately, they don't present any direct experimental evidence supporting this statement. Also, I'm unable to see in Fig. S1D that Vps13 foci form in proximity of ERMES, which was not labelled in this experiment. Moreover, the authors say that Gem1 deletion abrogated the punctate localization of Vps13. Unfortunately, they show only a single cell. More quantitative data are required here. Furthermore, Fig. S1D should include outlines of the cells that were imaged.

We have now compared Vps13^{GFP} foci to the *bona fide* MDC marker Tom70 (in this case mScarlet-tagged Tom70). Upon Mcp1 overexpression, Tom70-mScarlet signal is no longer homogeneously mitochondrial, but rather enriched in MDCs. Importantly, Vps13^{GFP} shares this feature, with Vps13^{GFP} foci colocalizing with Tom70-mScarlet foci, confirming that Vps13 localizes to MDCs. This data is now included in Supp. Figure 1E. To further support the *gem1* deletion data, we have updated the example in Figure 2 and added further examples (now Supplementary Figure 2). Finally, we added cell outlines for Supp. Fig. 1D.

4. The authors claim that ChiMERA rescues the growth defect of a *mdm34 mdm12* double mutant. This was tested by tetrad dissection. The authors show only two colonies with and two without the rescuing plasmid. More quantitative data are required to support this statement. The authors should perform drop dilution tests or obtain growth curves of at least three biological replicates.

This is a fair point that we have addressed by two different means. To address the quantification concern, we have added spot assays (now Figure 2G) for more quantitative comparison of ChiMERA-dependent growth rescue. To address the replication concern, we streaked three independent clones on non-fermentative media. ERMES deletion strains are not viable on this media whereas ChiMERA-expressing cells are. We see consistent rescue of growth in *mdm12mdm34* double delete cells expressing ChiMERA across the triplicate (Rebuttal Figure 1).

Rebuttal Figure 1: Streaks of three independent clones on YPG (non-fermentative media).

5. The authors compare rescue of *mdm10* delta by MMM1-mNG-FIS1C with rescue by ChiMERA (line 157). The lack of rescue by ChiMERA should be shown in the same experiment in Fig. 3D.

The rescue of *mdm10* by ChiMERA was already shown in the published literature (Kornmann et al. *Science* 325, 477-481 (2009), which is now cited in the text.

6. Do the mNG constructs used in Figs. 3 and 4 form punctate structures similar to native ERMES complexes? This could be easily tested by fluorescence microscopy.

We performed the suggested experiment to confirm that Mmm1-mNG-Fis1C formed dots like ERMES complexes (Supp. Fig. 4).

Reviewer #2 (Comments to the Authors (Required)):

The authors investigate how an artificial tether between ER and mitochondria (ChiMERA) without lipid transfer activity can rescue defects in the ER-mitochondria tether and lipid transfer ERMES complex. The authors present a systematic analysis of how different remaining ERMES subunits might work together to provide some residual function when ER and mitochondria are synthetically tethered. Using genetic approaches, the authors nicely demonstrate that Mmm1 provides sufficient remaining PL transfer activity to rescue the ERMES defect associated growth phenotype in artificial contexts of ChiMERA expression or

when itself being synthetically modified to increase membrane tethering.

This is a neat and focused study, but with likely marginal implications for a broader readership. Clearly, the proposed Mmm1-Mdm10 complex does not provide sufficient PL transfer activity under physiological conditions to promote mitochondria morphology or function without artificially stabilized ER-mitochondria tethering. However, differential residual activity in multicomponent complexes is not a new phenomenon and the actual biological implications in the wildtype setting remain unclear. The authors imagine the existence of ERMES complexes with differential subunit composition to for example bridge different distances of MCS, but do not provide any evidence for their actual existence in cells. Thus, the title of the manuscript implies cell biology not really shown in the manuscript.

To make the manuscript more appealing to a broader readership it would be important to (1) provide evidence for the existence and function of the proposed subcomplexes in wildtype cells or (2) substantiate the findings with an evolutionary link to other organisms with minimal ERMES complexes. Otherwise, the manuscript appears to address a rather technical or niche problem of unclear physiological relevance.

We agree with the reviewer that experimental evidence is lacking for the presence of ERMES variants with different subunit compositions in wild-type cells. Therefore, we have changed the title, summary, and discussion in the main text. In particular, we have changed the title to “Mitochondrially-tethered Mmm1 can function as a sole lipid transporter at ER-mitochondria contact sites” and added the following sentences to the text in the revised version; “Whether wild-type *S. cerevisiae* cells naturally contain a Mmm1–Mdm10 subcomplex remains unclear. If such forms exist, they are likely rare or transient, yet evidence from other species supports this two-protein model.”

As stated above, we do, however, disagree that our findings are only marginal and of limited interest. The idea that ERMES is important for lipid accumulation in mitochondria has retained contention since its discovery. First, ERMES is not essential in budding yeast, despite lipid transport to mitochondria being essential for the very existence of mitochondria, and second, ERMES depletion can be rescued by expression of a protein with no lipid transport activity. The discovery of the presence of a redundant lipid transfer pathway, namely Vps13-Mcp1, has addressed the former point, but why GFP glued between two membranes can rescue ERMES depletion has remained a stain on the model of ERMES' role in mitochondrial lipid accumulation. We explain this long-standing

discrepancy by showing that Mmm1 can function alone.

While we have not been able to demonstrate that wild-type cells contain a subpopulation of the minimal Mmm1-Mdm10 complex, as such a form is likely rare and/or transient and difficult to detect. Indeed, we failed in detecting a minor population of Mmm1-Mdm10 complex by crosslinking using a bifunctional crosslinker or benzoylphenylalanine site-specific photo-crosslinking in the absence or presence of ChiMERA (data not shown). However, earlier work in *S. pombe*, now seen in the light of our findings go a long way to demonstrate that, at least in some phylla, Mmm1 and Mdm10 can function by themselves without additional artificial tethering forces. (Rebuttal Figure 2).

Rebuttal Figure 2: Tetrad dissection in fission yeast showing Mmm1 overexpressing rescues ERMES depletion (taken from Li *et al.*, 2019. DOI: 10.1038/s41467-019-08928).

Crucially, Mmm1 overexpression (without any need for artificial tethering) can rescue *mdm12mdm34* double deletion but not *mdm10* deletion. This finding supports our Mmm1-Mdm10 dimer model. Rescue by Mmm1 overexpression does not work in budding yeast (Supp. Fig. 3B), highlighting evolutionary differences in the ERMES subcomplex stability and, potentially, importance.

Phylogenetic analysis also identifies a species that harbors only one ERMES member (Wideman *et al.*, 2013; <https://doi.org/10.1093/molbev/mst120>). Importantly, this is Mmm1, fitting our model. Nevertheless, given the limitations of such analyses, such as incomplete genomes and lack of downstream confirmation that these proteins are in fact localized at ER-mitochondria contact sites, we are careful not to overinterpret these data.

Reviewer #3 (Comments to the Authors (Required)):

The ER-mitochondria encounter structure ERMES consists of four obligatory components, deletion of each

resulting in a severe phenotype. ERMES is thought to assembly between the ER and mitochondrial membranes to channel lipids between the two organelles. An artificial tether without capacity to transfer lipids can rescue the deletion of the ERMES component Mdm12 or Mdm34, but deletions of Mdm10 or Mmm1 are only rescued to a low extent. This has been mysterious because Mdm12 and Mdm34 consist of similar lipid transfer domains as Mmm1.

This manuscript tests three possible models that could explain this mystery. Two models are ruled out experimentally, while the third one turns out to be the likely explanation. The authors conclude that of the four ERMES components, a minimal assembly of Mmm1 and Mdm10 is sufficient for function, provided that ER and mitochondria are tethered otherwise, such as through the artificial tether ChiMERA.

This study consists of a set of elegant, technically demanding and well carried-out experiments that shed new light on a long-standing question and is thus of wide interest to the cell biology community. I recommend it to be published, but have some suggestions for revisions as described below.

We would like to thank the reviewer for these positive comments.

Major:

Overall, the conclusions of this paper are well supported by the results. However, I find that the title, abstract and the introduction are somewhat misleading, as they imply that ERMES assemblies of variable composition do exist. While I agree with the authors that their results raise that possibility, it remains a hypothesis. There is no evidence that in wild type cell (i.e. in absence of ChiMERA expression) the Mmm1-Mdm10 complex makes up a significant fraction of lipid trajectories between the ER and mitochondria, or even exists.

We agree with the reviewer for this point. Please see our response to comment 1 by Reviewer 1.

In addition, the organisation of ERMES into the sequence Mmm1-Mdm12-Mdm34-Mdm10 is derived from biochemistry by Ellenrieder et al (Nat Comm 2021), not from cryotomography, which rather used the biochemistry results to interpret a low-resolution average structure. The statement in lines 44-49 of the introduction is thus a bit misleading, and Ellenrieder et al should be cited.

Thanks for pointing us towards the relevant literature, and away from the tendency to cite the latest

studies. We have cited Ellenrieder et al. (Nat Comm 2016) in the revised version.

Furthermore, the manuscript does not fully resolve the mystery of ChiMERA rescue, because the roles of Mdm12 and Mdm34 remain unaddressed. These two components do likely active lipid transfer domains, and their deletion growth phenotype is not fully rescued by ChiMERA to wild type level (Figure 2G in this manuscript and Kornmann et al 2009, Fig 2A), albeit the rescue is more pronounced than for Mdm10 and Mmm1. Thus, they must have a function beyond only tethering.

We apologies for the lack of clarity here. We do indeed believe that Mdm12 and Mdm34 have a role in lipid transport and therefore beyond tethering (Wild type model in Figure 1). Our conclusions are with regards to why the single deletion of Mmm1 (i.e., in the presence of Mdm12 and Mdm34) cannot be rescued by artificial tethering. For further clarity, we have added the following sentence to the manuscript, “*This Model 3 explains why ChiMERA expression fails to efficiently rescue loss of Mmm1 or Mdm10, in contrast to deletions of Mdm12 or Mdm34 and highlight a special role for Mmm1 and Mdm10 that cannot be substituted by tethering forces alone. Though in heterotetrameric ERMES complexes, Mdm12 and Mdm34 are still involved in efficient transport of lipids by organizing multiple SMP domain in tandem.*”

Minor:

Lines 138-139 (and other instances) I do not think the term 'redundant' is used correctly regarding Mdm12 and Mdm34. The authors mean to say that Mdm12 and Mdm34 cannot replace each other (and I agree based on their results), but redundant can also mean dispensable; thus not needed or superfluous. In that sense, saying that Mdm12 and Mdm34 are not redundant seems misleading with regard to the results. The authors might want to reconsider the use of the term redundant.

We agree the use of “redundancy” in the original manuscript is not always clear as to whether it is referring to the redundancy between Mdm12 or Mdm34 or just that Mdm12 and Mdm34 are dispensable in general. We have now modified the text to more accurately reflect that the redundancy we are referring to is between the two proteins in lipid transport.

Lines 90-97: It is unclear whether the authors mean to say that these Vps13 spots appear near ERMES in wild type cells or when Mcp1 is overexpressed.

This result refers to Mcp1-overexpressing cells. We have updated the main text to make this clearer.

Figure 1 is perhaps not necessary. The key panels are shown again in Figures 2 and 3.

Since this figure summarizes the models to be tested in this study, we think it is still helpful for readers. We have therefore retained it as it is.

Line 143: "providing tethering force" The authors might want to reconsider this phrasing; there is no indication that Mdm12 or Mdm34 are able to provide any force.

We agree with the reviewer on this point and have thus altered the sentence to “*Contrary to deficiencies in Mdm12, Mdm34 and combination thereof, deficiency in Mmm1 and Mdm10 are poorly, if at all, rescued by ChiMERA expression (Kornmann et al., 2009), indicating that these components play a special role beyond bridging the two membrane, a function that ChiMERA can fulfill.*”

Line 148: "...gap induced by" It is not known what gap ChiMERA mediated tethering induces; what the authors mean is the predicted gap. It has not been experimentally determined.

We have altered the sentence to “*In fact, AlphaFold3 predicts a highly confident Mmm1-Mdm10 heterodimer in an orientation compatible with lipid transfer between two membranes and of an appropriate size to fit within the close apposition between the ER and mitochondria, predicted to be induced by ChiMERA-mediated tethering (Figure 3B).*”

Line 117: Typo, components should be component

We have corrected the text accordingly. Thanks!

Line 471 : Typo, mdm34-mCherry should be Mdm34-mCherry

We have corrected the text accordingly. Thanks!

October 22, 2025

Re: JCB manuscript #202411196R

Benoit Kornmann
University of Oxford

Dear Dr. Kornmann,

Thank you for submitting your revised manuscript entitled "Mitochondrially-tethered Mmm1 can function as sole lipid transporter at ER-mitochondria contact sites." The manuscript has been seen by two of the original reviewers whose full comments are appended below. You will see that while one reviewer is positive about the work in terms of its suitability for JCB, the second reviewer raises important issues that would need to be addressed prior to acceptance.

The current data shows that Mmm1 alone is able to substitute for ERMES but it remains open whether this is due to membrane tethering (as done by an artificial tether) or by lipid transfer. Therefore, as suggested by the reviewer, testing one of the described point mutations in Mmm1 that inhibit lipid transfer would unambiguously clarify the effect of the artificial tether. Understanding the differences between SMP domains and characterization of the predicted interface between Mmm1 and Mdm10 are interesting and important questions but seem to us to be beyond the scope of this paper. However, please address these and the other minor points with changes to text and figures.

Our general policy is that papers are considered through only one revision cycle. However, in this case we are open to one additional short round of revision that addresses the remaining concerns. Please submit the final revision along with a cover letter that includes a point by point response to the reviewer comments.

Thank you for this interesting contribution to Journal of Cell Biology. You can contact me or the scientific editor listed below at the journal office with any questions at cellbio@rockefeller.edu.

Sincerely,

Thomas Langer, PhD
Monitoring Editor
Journal of Cell Biology

Dan Simon, PhD
Scientific Editor
Journal of Cell Biology

Reviewer #1 (Comments to the Authors (Required)):

The authors have responded to my previous concerns in an adequate manner. Publication of this interesting manuscript can now be recommended. Only a very minor point remains that should be corrected before publication.

1. I understand that Figure 2C shows the proportion of Vps13 foci colocalizing with Mdm34, rather than absolute numbers. The labeling of the y-axis should be adapted accordingly.

Reviewer #2 (Comments to the Authors (Required)):

In this revised/resubmitted manuscript, the authors re-emphasized their findings away from the speculated existence of minimal Mmm1-Mdm10 complexes towards the central role of Mmm1 as a sole lipid transporter when synthetically tethered to mitochondria.

The controversy appeared to be in 2012 by Nguyen et al and to some extent by Voss et al. However, as the authors state, the controversy of whether ERMES functions as a lipid transport protein had been largely resolved in the following years by structural, biochemical and in vitro lipid transport approaches. Thus, to me, the question of how the artificial tether ChiMERA contributes to the rescue of ERMES defects is a more historical/technical issue of interest to ERMES specialists, unless it revealed some new biology. The fact that Mmm1 or its SMP domain, when artificially tethered to ER and mitochondria, can rescue the growth phenotype associated with the loss of ERMES, is interesting and suggests a more important or potentially differential role for Mmm1 within ERMES compared with Mdm12/34 for lipid transport between ER and mitochondria.

Unfortunately, the manuscript does not provide further insights into this point. It would have been very interesting to explore the molecular basis for the differences in their ability to rescue growth between the SMP domains from Mmm1, Mdm12 or Mdm34. Is there something special about the SMP domain of Mmm1 or are the differences caused by the specific context of the synthetic constructs? To test the role of Mmm1-mediated lipid transport in the context of the assembled ERMES, it would have been important to test what happens in the presence of Mdm12 and/or Mdm34 variants, which assemble but cannot transport lipid due to specific mutations. Could their presence be required to support stable association of Mmm1 and Mdm10 rather than functioning as lipid transporter themselves. Or does their transport function in parallel to or in conjunction with Mmm1 significantly contribute to ERMES function. Thus, to me, the significance of the finding that Mmm1 when artificially tethered to mitochondria can at least minimally replace fully assembled ERMES in terms of cell growth and mitochondrial morphology is a bit underwhelming in a physiological context. As the authors showed in the past, directing a high copy number of Vps13 to mitochondria can rescue the loss of ERMES function. Thus, could any or many artificial transport constructs unrelated to ERMES components compensate for the loss of ERMES and the artificial tether of Mmm1 being one of them? To address the controversy more directly, it would have been interesting/important to analyze the potential interface between Mmm1 and Mdm10 proposed by AlphaFold modeling. Currently, Model 3 is not directly experimentally tested by the authors. So, the mystery of ChiMERA remains only indirectly answered.

Given the (re-)emphasis on resolving the discussion between the tethering and lipid transport function of ERMES, key experiments are missing from the revised/resubmitted manuscript. The authors do not provide evidence that it is the lipid transport function of Mmm1 that is required for the rescue. Yes, the full-length Mmm1 or only its SMP domain when synthetically tethered to the ER and mitochondria can rescue. While likely, this, however, does not prove that it is based on its lipid transport activity. Thus, a crucial experiment is to express versions of the different rescue variants which carry point mutations specifically preventing lipid binding/transport but not membrane tethering. Possibly beyond the scope of the manuscript, but to assess the role of Mmm1 as a sole lipid transporter, it would be informative to measure lipid transport from the ER to mitochondria directly and quantitatively. Currently, we can only conclude that the different synthetic tethers seem to restore at least minimal transport activity (pending analysis above) supporting cell growth and mitochondrial morphology (which should be shown/analyzed for all rescue constructs).

Minor points:

The authors need to show the controls for the protein levels of the different constructs used in figures 3 and 4 to confirm that these are not factors in the ability to rescue the tested phenotypes.

Dear Editor, dear Thomas,

Thank you for handling our manuscript and setting clear and addressable revision targets. In response to Reviewer #2, we have set out to address whether or not it is indeed the lipid transport function of mitochondrially tethered Mmm1, rather than tethering alone, that rescues the complete depletion of all ERMES subunits. This has been achieved by two means; firstly, we have now shown that artificially tethered Mmm1 rescues the aberrant cardiolipin levels found in ERMES depleted cells, highlighting not just a rescue in growth and mitochondrial morphology, but importantly of the lipidome; and secondly, informed by published crystal structures, we used molecular simulation-based mutagenesis tools to design and express a stable construct with key lipid-binding hydrophobic residues mutated to hydrophilic, thus preventing lipid transport without impairing protein folding and expression. This quintuple point mutant of Mmm1 does not rescue the growth upon loss of Mmm1, despite tethering the ER and mitochondria.

We believe that these data, in conjunction with the variety of artificial tether constructs included in prior version of the manuscript that do not rescue full ERMES depletion (e.g., ChiMERA, Sec66(TM)-Mdm12-Fis1(TM), Sec66(TM)-Mdm34-Fis1(TM), and various truncations of tethered Mmm1), definitely show that tethering alone does not rescue ERMES deficiency. Instead, rescue is obtained by the anchoring of Mmm1's lipid transporting SMP domain to ER-mitochondria contacts, demonstrating that it can function as the sole lipid transporter at ER-mitochondria contact sites.

With our best wishes,

Benoit Kornmann
Toshi Endo

Reviewer #1 (Comments to the Authors (Required)):

The authors have responded to my previous concerns in an adequate manner. Publication of this interesting manuscript can now be recommended. Only a very minor point remains that should be corrected before publication.

1. I understand that Figure 2C shows the proportion of Vps13 foci colocalizing with Mdm34, rather than absolute numbers. The labeling of the y-axis should be adapted accordingly.

Thank you for this positive evaluation. The proposed change is indeed important and the label has been changed to “proportion of Vps13 foci with Mdm34”

Reviewer #2 (Comments to the Authors (Required)):

In this revised/resubmitted manuscript, the authors re-emphasized their findings away from the speculated existence of minimal Mmm1-Mdm10 complexes towards the central role of Mmm1 as a sole lipid transporter when synthetically tethered to mitochondria.

The controversy appeared to be in 2012 by Nguyen et al and to some extent by Voss et al. However, as the authors state, the controversy of whether ERMES functions as a lipid transport protein had been largely resolved in the following years by structural, biochemical and in vitro lipid transport approaches. Thus, to me, the question of how the artificial tether ChiMERA contributes to the rescue of ERMES defects is a more historical/technical issue of interest to ERMES specialists, unless it revealed some new biology. The fact that Mmm1 or its SMP domain, when artificially

tethered to ER and mitochondria, can rescue the growth phenotype associated with the loss of ERMES, is interesting and suggests a more important or potentially differential role for Mmm1 within ERMES compared with Mdm12/34 for lipid transport between ER and mitochondria. Unfortunately, the manuscript does not provide further insights into this point. It would have been very interesting to explore the molecular basis for the differences in their ability to rescue growth between the SMP domains from Mmm1, Mdm12 or Mdm34. Is there something special about the SMP domain of Mmm1 or are the differences caused by the specific context of the synthetic constructs? To test the role of Mmm1-mediated lipid transport in the context of the assembled ERMES, it would have been important to test what happens in the presence of Mdm12 and/or Mdm34 variants, which assemble but cannot transport lipid due to specific mutations. Could their presence be required to support stable association of Mmm1 and Mdm10 rather than functioning as lipid transporter themselves. Or does their transport function in parallel to or in conjunction with Mmm1 significantly contribute to ERMES function. Thus, to me, the significance of the finding that Mmm1 when artificially tethered to mitochondria can at least minimally replace fully assembled ERMES in terms of cell growth and mitochondrial morphology is a bit underwhelming in a physiological context. As the authors showed in the past, directing a high copy number of Vps13 to mitochondria can rescue the loss of ERMES function. Thus, could any or many artificial transport constructs unrelated to ERMES components compensate for the loss of ERMES and the artificial tether of Mmm1 being one of them?

To address the controversy more directly, it would have been interesting/important to analyze the potential interface between Mmm1 and Mdm10 proposed by AlphaFold modeling. Currently, Model 3 is not directly experimentally tested by the authors. So, the mystery of ChiMERA remains only indirectly answered.

Given the (re-)emphasis on resolving the discussion between the tethering and lipid transport function of ERMES, key experiments are missing from the revised/resubmitted manuscript. The authors do not provide evidence that it is the lipid transport function of Mmm1 that is required for the rescue. Yes, the full-length Mmm1 or only its SMP domain when synthetically tethered to the ER and mitochondria can rescue. While likely, this, however, does not prove that it is based on its lipid transport activity. Thus, a crucial experiment is to express versions of the different rescue variants which carry point mutations specifically preventing lipid binding/transport but not membrane tethering. Possibly beyond the scope of the manuscript, but to assess the role of Mmm1 as a sole lipid transporter, it would be informative to measure lipid transport from the ER to mitochondria directly and quantitatively. Currently, we can only conclude that the different synthetic tethers seem to restore at least minimal transport activity (pending analysis above) supporting cell growth and mitochondrial morphology (which should be shown/analyzed for all rescue constructs).

To address whether it is indeed the lipid transport function of Mmm1 that is required for the rescue, and following guidance from the editor, we tested the two following predictions; 1st, if Mmm1 rescued via its lipid transport activity, this should show on the lipidome of cells rescued by tethered Mmm1. We therefore addressed the most severe lipid defect associated with ERMES deficiency, i.e. the strong reduction of cardiolipin accumulation, and found that this lipidomic defect was indeed rescued by the expression of tethered Mmm1 (Fig. 4C and D in the revised manuscript). 2nd and most important, we directly tested whether Mmm1's ability to rescue was based on lipid transport activity by generating a lipid-binding impaired Mmm1 mutant and found that lipid binding was necessary for rescue to take place. Of note, previously designed single-mutants of Mmm1 had only modest effects on its lipid-transport ability, and we found that they rescue growth like wild-type (Kawano et al., JCB 2018). To generate a bona fide lipid transport mutant informed by published structures of Mmm1, we used molecular simulation-assisted mutagenesis to generate a quintuple mutant that opens a hydrophilic wedge within the hydrophobic groove of Mmm1, but crucially, does not destabilise the protein (Fig. 5C in the revised manuscript). We observe that this protein is

expressed as robustly as its wild-type counterpart (Fig. 5D in the revised manuscript) and localized at the ER-mitochondrial contacts (Supp. Fig. 4, bottom panels in the revised manuscript), but fails to rescue growth (Fig. 5E in the revised manuscript), validating the idea that lipid binding and transport by Mmm1 is necessary for rescue.

Minor points:

The authors need to show the controls for the protein levels of the different constructs used in figures 3 and 4 to confirm that these are not factors in the ability to rescue the tested phenotypes.

This has been done and added to Sup. Figure 5. Note that the proteins express well, with their level showing no correlation to their ability to rescue growth. One protein (Sec66N-MDM12-3FLAG-mNG-Fis1C) could not be detected when expressed from its endogenous promoter, but the similar FLAG-less Sec66N-MDM12-mNG-Fis1C construct expressed from the ADH1 promoter, while very well expressed, fails nonetheless to rescue growth.

March 26, 2026

RE: JCB Manuscript #202411196RR

Benoit Kornmann
University of Oxford

Dear Dr. Kornmann,

Thank you for submitting your revised manuscript entitled "Mitochondrially-tethered Mmm1 can function as sole lipid transporter at ER-mitochondria contacts." We would be happy to publish your paper in JCB pending final revisions necessary to meet our formatting guidelines (see details below).

A. MANUSCRIPT ORGANIZATION AND FORMATTING:

1) Text limits: Character count for Reports is < 20,000, not including spaces. Count includes title page, abstract, introduction, results & discussion, and acknowledgments. Count does not include materials and methods, figure legends, references, tables, or supplemental legends.

**** Reports must have a single 'Results and Discussion' section. ****

2) Figure formatting: Reports may have up to 5 main text figures. Scale bars must be present on all microscopy images, including inset magnifications. Molecular weight or nucleic acid size markers must be included on all gel electrophoresis. Please add scale bars to figures 4B, S1B, & S2 and MW markers to figures 5D & S5(Tom70 blots). Size markers on gels and blots cannot be the expected sizes of the proteins of interest. In order for readers to accurately assess the size of the proteins shown, the cropped images must extend to include a region containing at least one of the molecular weight size markers that were run on the gel.

Also, please avoid pairing red and green for images and graphs to ensure legibility for color-blind readers. If red and green are paired for images, please ensure that the particular red and green hues used in micrographs are distinctive with any of the colorblind types. If not, please modify colors accordingly or provide separate images of the individual channels.

3) Statistical analysis: Error bars on graphic representations of numerical data must be clearly described in the figure legend. The number of independent data points (n) represented in a graph must be indicated in the legend. Please indicate whether 'n' refers to technical or biological replicates (i.e. number of analyzed cells, samples or animals, number of independent experiments). If independent experiments with multiple biological replicates have been performed, we recommend using distribution-reproducibility SuperPlots (please see Lord et al., JCB 2020) to better display the distribution of the entire dataset, and report statistics (such as means, error bars, and P values) that address the reproducibility of the findings.

Statistical methods should be explained in full in the materials and methods. For figures presenting pooled data the statistical measure should be defined in the figure legends. Please also be sure to indicate the statistical tests used in each of your experiments (both in the figure legend itself and in a separate methods section) as well as the parameters of the test (for example, if you ran a t-test, please indicate if it was one- or two-sided, etc.). Also, if you used parametric tests, please indicate if the data distribution was tested for normality (and if so, how). If not, you must state something to the effect that "Data distribution was assumed to be normal but this was not formally tested."

4) Materials and methods: Should be comprehensive and not simply reference a previous publication for details on how an experiment was performed. Please provide full descriptions (at least in brief) in the text for readers who may not have access to referenced manuscripts. The text should not refer to methods "...as previously described." Please also indicate the acquisition and quantification methods for immunoblotting/western blots.

5) Since your paper contains AlphaFold modeling please include PAE and pLDDT quality control metrics for all models.

6) For all cell lines, vectors, strains, constructs/cDNAs, etc. - all genetic material: please include database / vendor ID (e.g. Addgene, ATCC, etc.) or if unavailable, please briefly describe their basic genetic features, even if described in other published work or gifted to you by other investigators (and provide references where appropriate). Please be sure to provide the sequences for all of your oligos: primers, si/shRNA, RNAi, gRNAs, etc. in the materials and methods. You must also indicate in the methods the source, species, and catalog numbers/vendor identifiers (where appropriate) for all of your antibodies, including

secondary. If antibodies are not commercial, please add a reference citation if possible.

7) Microscope image acquisition: The following information must be provided about the acquisition and processing of images:

- a. Make and model of microscope
- b. Type, magnification, and numerical aperture of the objective lenses
- c. Temperature
- d. Imaging medium
- e. Fluorochromes
- f. Camera make and model
- g. Acquisition software
- h. Any software used for image processing subsequent to data acquisition. Please include details and types of operations involved (e.g., type of deconvolution, 3D reconstitutions, surface or volume rendering, gamma adjustments, etc.).

8) References: There is no limit to the number of references cited in a manuscript. References should be cited parenthetically in the text by author and year of publication. Abbreviate the names of journals according to PubMed.

9) Supplemental materials: Reports typically have up to 3 supplemental figures. You currently exceed this limit but, in this case, we will be able to give you the extra space but please try not to add to the current total. Please also note that tables, like figures, should be provided as individual, editable files. A summary of all supplemental material should appear at the end of the Materials and methods section. Please include one brief sentence per item.

10) eTOC summary: A ~40-50 word summary that describes the context and significance of the findings for a general readership should be included on the title page. The statement should be written in the present tense and refer to the work in the third person. It should begin with "First author name(s) et al..." to match our preferred style.

11) Conflict of interest statement: JCB requires inclusion of a statement in the acknowledgements regarding competing financial interests. If no competing financial interests exist, please include the following statement: "The authors declare no competing financial interests." If competing interests are declared, please follow your statement of these competing interests with the following statement: "The authors declare no further competing financial interests."

12) A separate author contribution section is required following the Acknowledgments in all research manuscripts. All authors should be mentioned and designated by their first and middle initials and full surnames. We encourage use of the CRediT nomenclature (<https://casrai.org/credit/>).

13) ORCID IDs: ORCID IDs are unique identifiers allowing researchers to create a record of their various scholarly contributions in a single place. Please note that ORCID IDs are required for all authors. At resubmission of your final files, please be sure to provide your ORCID ID and those of all co-authors.

14) JCB requires authors to submit Source Data used to generate figures containing gels and Western blots with all revised manuscripts. This Source Data consists of fully uncropped and unprocessed images for each gel/blot displayed in the main and supplemental figures. For assays performed using capillary electrophoresis and/or immunoassay-based detection, authors should instead provide the electropherogram graph(s) for each experiment, plotting fluorescence/chemiluminescence intensity vs. molecular weight/size. Since your paper includes cropped gel and/or blot images, please be sure to provide one Source Data file for each figure gels, blots, and/or capillary electrophoresis assays along with your revised manuscript files. File names for Source Data figures should be alphanumeric without any spaces or special characters (i.e., SourceDataF#, where F# refers to the associated main figure number or SourceDataFS# for those associated with Supplementary figures). For traditional gels and blots, the lanes of the gels/blots should be labeled as they are in the associated figure, the place where cropping was applied should be marked (with a box), and molecular weight/size standards should be labeled wherever possible. For capillary electrophoresis assays, each trace in the graph should be color-coded and labeled to indicate which protein, gene, or sample is being measured (please try to avoid red/green combinations to accommodate our color-blind readers).

Source Data files will be directly linked to specific figures in the published article. Source Data Figures should be provided as individual PDF files (one file per figure). Authors should endeavor to retain a minimum resolution of 300 dpi or pixels per inch. Please review our instructions for export from Photoshop, Illustrator, and PowerPoint here: <https://rupress.org/jcb/pages/submission-guidelines#revised>.

15) Journal of Cell Biology now requires a data availability statement for all research article submissions. These statements will be published in the article directly above the Acknowledgments. The statement should address all data underlying the research presented in the manuscript. Please visit the JCB instructions for authors for guidelines and examples of statements at (<https://rupress.org/jcb/pages/editorial-policies#data-availability-statement>).

B. FINAL FILES:

Please upload the following materials to our online submission system. These items are required prior to acceptance. If you

have any questions, contact JCB's Managing Editor, Lindsey Hollander (lhollander@rockefeller.edu).

****It is JCB policy that if requested, original data images must be made available to the editors. Failure to provide original images upon request will result in unavoidable delays in publication. Please ensure that you have access to all original data images prior to final submission.****

****The license to publish form must be signed before your manuscript can be sent to production. A link to the license to publish form will be sent to the corresponding author only. Please take a moment to check your funder requirements before choosing the appropriate license.****

Thank you for your attention to these final processing requirements. Please revise and format the manuscript and upload materials within 7 days. If you need an extension for whatever reason, please let us know and we can work with you to determine a suitable revision period.

Thank you for this interesting contribution, we look forward to publishing your paper in Journal of Cell Biology.

Sincerely,

Thomas Langer, PhD
Monitoring Editor
Journal of Cell Biology

Dan Simon, PhD
Scientific Editor
Journal of Cell Biology